Corrected: Publisher correction

# Cytoplasmic localization of GRHL3 upon epidermal differentiation triggers cell shape change for epithelial morphogenesis

Chiharu Kimura-Yoshida [1], Kyoko Mochida[1], Masa-aki Nakaya[2], Takeomi Mizutani[3] & Isao Matsuo [1]

Epithelial cell shape change is a pivotal driving force for morphogenesis of complex three-dimensional architecture. However, molecular mechanisms triggering shape changes of epithelial cells in the course of growth and differentiation have not been entirely elucidated. *Grhl3* plays a crucial role as a downstream transcription factor of Wnt/$\beta$-*catenin* in epidermal differentiation. Here, we show *Grhl3* induced large, mature epidermal cells, enriched with actomyosin networks, from embryoid bodies in vitro. Such epidermal cells were apparently formed by the simultaneous activation of canonical and non-canonical Wnt signaling pathways. A nuclear transcription factor, GRHL3 is localized in the cytoplasm and cell membrane during epidermal differentiation. Subsequently, such extranuclear GRHL3 is essential for the membrane-associated expression of VANGL2 and CELSR1. Cytoplasmic GRHL3, thereby, allows epidermal cells to acquire mechanical properties for changes in epithelial cell shape. Thus, we propose that cytoplasmic localization of GRHL3 upon epidermal differentiation directly triggers epithelial morphogenesis.

[1] Department of Molecular Embryology, Research Institute, Osaka Women's and Children's Hospital, Osaka Prefectural Hospital Organization 840, Murodo-cho, Izumi, Osaka 594-1101, Japan. [2] Department of Molecular Biology, Yokohama City University Graduate School of Medicine, 3-9 Fukuura, Kanazawa-ku, Yokohama, 236-0004 Kanagawa, Japan. [3] Department of Life Science and Technology, Faculty of Engineering, Hokkai-Gakuen University, Nishi 11-chome, Minami 26-jo, Chuo-ku, Sapporo, Hokkaido 064-0926, Japan. Correspondence and requests for materials should be addressed to C.K-Y. (email: chiharu@wch.opho.jp) or to I.M. (email: imatsuo@wch.opho.jp)

Epithelial morphogenesis directed by multiple cellular processes, such as cell shape changes, proliferation, and migration, involves the formation of complex three-dimensional architecture as seen, for example, in the formation of the mammalian neural tube[1]. Coordinated cell shape changes, including contraction and elongation along the apical–basal axis, play a pivotal role in epithelial morphogenesis[2,3]. Such cell deformations are controlled by cellular mechanical stress and tension at the cell surface primarily through intracellular contractile actomyosin networks[1,4–7]. However, the mechanisms initiating the morphogenesis of epithelial cells in coordination with the timing of specification, i.e. cell growth or differentiation during development, are relatively unknown.

The epidermis constitutes the outermost epithelial layer that wraps the entire body and changes the body form. During epidermal development, the single ectodermal sheet is largely specified into neural and surface ectoderm (SE), which is an immature embryonic state of epidermis[8]. Such temporal SE cells subsequently commit into periderm and finally form mature epidermis, the outer component of the skin[8]. Epidermal specification is initiated by the signaling of several growth factors. Bone morphogenetic protein signaling has been shown to direct epidermal specification in frog and zebrafish embryos[9]. Wnt signaling has also been suggested to be involved in epidermal specification in chick and mouse embryos[10–13]. Notably, we have found that during neural tube closure the canonical Wnt signaling pathway progressively specifies SE fate at the neural plate border, where neither surface nor neural cells are specified as uncommitted ectodermal progenitors[14]. These findings have led to the hypothesis that cell fate specification of SE during neurulation may be intimately linked to the epithelial morphogenesis of primary neurulation, which is governed by the non-canonical Wnt pathway involving planar cell polarity (PCP) genes[15]. However, little is known about what, when, and how molecular mechanisms control the coordination of epidermal fate decision and PCP-mediated epithelial morphogenesis.

The Grainy head family of transcription factors plays a highly conserved role in epithelial tissue development and remodeling in the animal kingdom[16,17]. Since epithelial morphogenesis is one of the major driving forces of neurulation[18–20], the mammalian Grainy head family, encoding Grainyhead-like (Grhl) factors, is crucial for neural tube formation[21,22]. During mouse primary neurulation, Grhl3 acts as a downstream effector of Wnt/β-catenin signaling to direct the specification of SE, a developmentally temporal structure of epidermis[8,14,23,24]. However, molecular mechanisms underlying Grhl3-dependent epidermal differentiation in any subsequent epithelial morphogenesis remain poorly understood.

In this study, we demonstrate how the differentiation of epidermal cells intimately links to cell shape changes through Grhl3. Mechanistically, GRHL3, which has been considered as a transcription factor, appears to localize in the cytoplasm upon differentiation over time to activate non-canonical Wnt signaling. Such GRHL3-induced epidermal cells acquire mechanical properties necessary for cell shape change through actomyosin networks. This finding provides unique insights into the temporal mechanisms involved in the initiation of epithelial morphogenesis in coordination with cell specification during animal development.

## Results

**Grhl3 induces large and mature epidermal cells.** During primary neurulation, Grhl3 can specify cellular fate into SE from ectodermal progenitor cells in neural folds[14]. To analyze the precise molecular mechanisms underlying epidermal differentiation by Grhl3, a two-step in vitro culture system was exploited (Fig. 1a)[25]. Embryonic stem (ES) cell-derived embryoid bodies (EBs) consisted of central undifferentiated progenitor cells and TROMA-1 (keratin 8) positive peripheral epidermal cells, which are formed without any additional factors (Fig. 1b). In this system, the overexpression of Grhl3 cDNA induced epidermal cells within central EBs that were distinct from those in the periphery of EBs by control vector (pgk-neo) transfectants (Fig. 1b, c, g; n = 33/33 100%). To explore the role of Grhl3 in more detail, Grhl3-induced epidermal cells were characterized (Fig. 1h, i, Supplementary Fig. 1a–g); histological examination revealed these to be large and multinucleated (Fig. 1h, i). Molecular analyses using epidermal markers indicated that such large multinucleated epidermal cells were more mature than epidermal cells in the periphery of EBs (Supplementary Fig. 1a–h). In this regard, a marker for the neural crest, SLUG, was not expressed in large, mature (LM)-epidermal cells (Supplementary Fig. 1i, j). Furthermore, LM-epidermal cells were highly enriched with actomyosin as well as keratin bundles (Supplementary Fig. 1g, l, m). These findings indicate that Grhl3 can induce distinct LM-epidermal cells in the central region of EBs.

To identify whether Grhl1, another member of the Grhl family, could induce LM-epidermal cells in a similar manner, we overexpressed Grhl1 cDNA in EBs and found that Grhl1 did not induce LM-epidermal cells efficiently (Fig. 1d, g). However, Grhl1 cDNA appeared to induce defective types of LM-epidermal cells, designated as "solitary" and "scattered" epidermal cells, in EBs (Supplementary Fig. 1n, o). Afterward, we defined these two epidermal cell types cytomorphologically as follows: "solitary" epidermal cells consisted of a single epidermal cell in isolation but not multinucleated in EBs, while "scattered" epidermal cells formed as aggregates in EBs but not in multinucleated cells (Supplementary Fig. 1n, o). "None" means that TROMA-1–positive epidermal cells were found in EBs but not in the periphery (Supplementary Fig. 1k, p, Fig. 1b). Concurrently, a dominant-negative form of Grhl3 cDNA (DN-Grhl3), which can inhibit the activity of the Grhl family[26], reduced the number of LM-epidermal cells induced by Grhl3 cDNA (Fig. 1e–g). These findings indicate that Grhl3 has a distinct role in inducing "LM-epidermal cells" in EBs in vitro.

To determine whether Grhl3 could also induce LM-epidermal cells in vivo, we generated transgenic mice that misexpressed Grhl3 cDNA (Fig. 2a, b). To create Grhl3-misexpressing embryos, we crossed transgenic mice carrying CAG-loxP-lacZ-loxP-Grhl3 (CAG-lacZ-Grhl3) cDNA, in which the lacZ gene is flanked by two loxP sites and the Grhl3 cDNA is under the control of the CAG promoter, and β-actin cre transgenic mice (Tg(CAG-Grhl3); Fig. 2a, b). Morphological abnormalities in Tg(CAG-Grhl3) embryos were observed from E4.5 to E5.5 (Fig. 2c–f). Tg(CAG-Grhl3) embryos also failed to develop and grow after implantation (Fig. 2d, f). To explore the possibility that Grhl3 may have induced actomyosin enrichment and multinucleation, we analyzed actin and nuclear markers in Tg(CAG-Grhl3) embryos (Fig. 3g–j). Indeed, F-actin expression was drastically enriched in the cell membrane of the Tg(CAG-Grhl3) embryos (Fig. 2h; n = 3). Additionally, some cells became multinucleated (Fig. 2h, j; white arrows (n = 5,3, respectively)). Moreover, TROMA-1 expression appeared to be enhanced in Tg(CAG-Grhl3) embryos (Fig. 2l; n = 4). These findings suggest that Grhl3 misexpression promotes multinucleation and F-actin enrichment, being comparable to LM-epidermal cells derived from EBs in vitro (Fig. 1)

Furthermore, we misexpressed Grhl3 cDNA locally in SE cells during neurulation and analyzed phenotypes (Fig. 2m–u). Medium containing CAG-Grhl3 cDNA and CAG-EGFP plasmids was injected into the wild-type exocoelomic cavity at E8.5 (Fig. 2m) and, thereafter, the plasmids were electroporated into

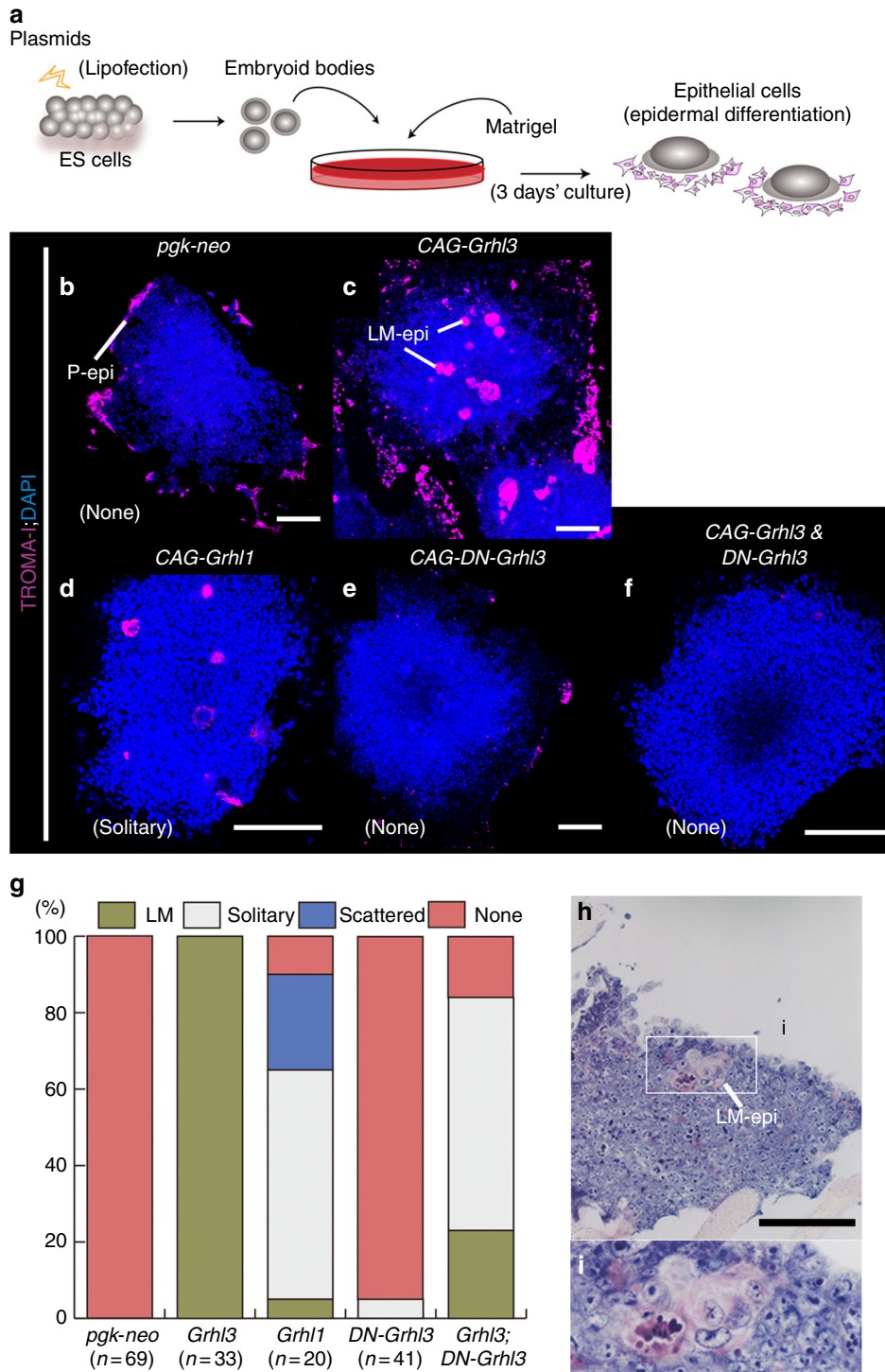

**Fig. 1** *Grhl3* induces large and mature epidermal cells from embryoid bodies in vitro. **a** Schematic protocol for inducing epidermal cells from embryoid bodies (EBs). Initially, cystic EBs developed from dissociated embryonic stem (ES) cells at high density on an uncoated dish. Then, EB aggregates were cultured on a Matrigel-coated dish and assessed for their ability to progress along epithelial lineages. **b–f** Marker expression analysis in differentiated epidermal cells. TROMA-1 (magenta) and DAPI (blue). ES cells were transfected with control vector *pgk-neo* (**b**), *CAG-Grhl3* (**c**), *CAG-Grhl1* (**d**), *CAG-DN-Grhl3* (*dominant-negative Grhl3*) (**e**), or *CAG-Grhl3/CAG-DN-Grhl3* (**f**). LM-epi: large and mature epidermal cells found in the central region of EBs (**c**). P-epi: peripheral epidermal cells found in the periphery or outside of the EBs (**b**). **g** Frequency of epidermal cells among EBs induced by *pgk-neo*, *Grhl3* cDNA, *Grhl1* cDNA, or *CAG-Grhl3/DN-Grhl3* are represented. TROMA-I–positive central epidermal cells were classified into three types: LM-, solitary and scattered epidermal cells. "*n*" indicates the number of EBs analyzed. **h, i** Morphological features of large and mature (LM) epidermal cells using hematoxylin–eosin staining. Representative images and frequency from more than three independent experiments. Scale bars represent 200 μm (**b–f**, **h**)

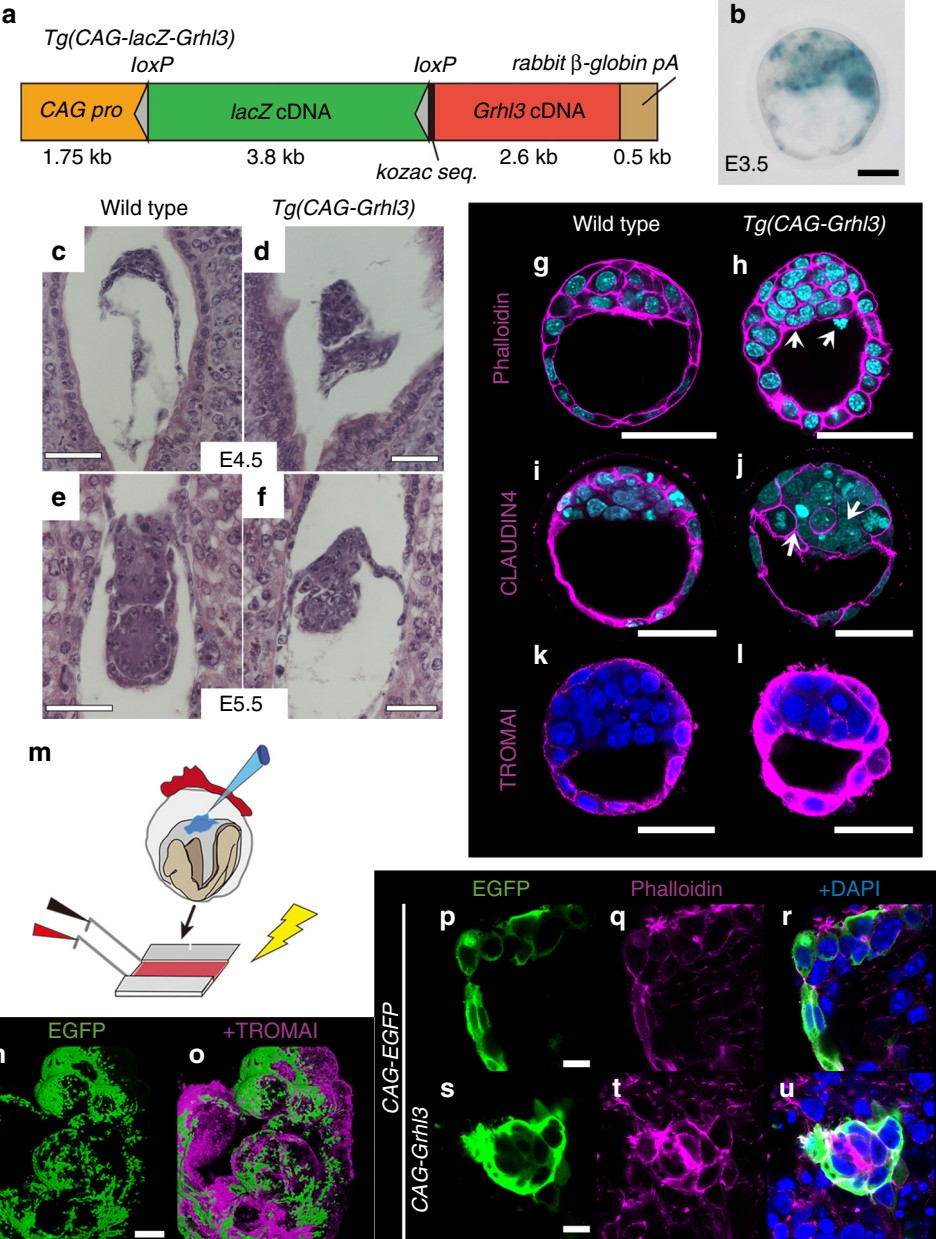

**Fig. 2** *Grhl3* promotes cytoskeletal networks and multinucleation in vivo. **a** A schematic construct of the *CAG-loxP-lacZ-loxP-Grhl3 (CAG-lacZ-Grhl3)* transgene. **b** *LacZ* expression in the *Tg(CAG-lacZ-Grhl3)* blastocyst with X-gal staining. **c–f** Morphological features of wild type (**c**, **e**) and *Tg(CAG-Grhl3)* (**d**, **f**) at E4.5 (**c**, **d**), and E5.5 (**e**, **f**), respectively. Hematoxylin–eosin sagittal sections. **g–l** Immunohistochemical analyses of whole embryos at E3.5 in wild type (**g**, **i**, **k**) and *Tg (CAG-Grhl3)* (**h**, **j**, **l**). Phalloidin (magenta; **g**, **h**), CLAUDIN4 (magenta; **i**, **j**) and TROMA-1 (magenta; **k**, **l**), and DAPI (cyan; **g–j**, blue; **k**, **l**) stains. Multinucleate cells are seen in the inner cell mass region (arrows; **h**, **j**). **m** Schematic protocol for the electroporation of vectors, *CAG-EGFP* with or without *CAG-Grhl3*, into the embryo proper at E8.5. **n**, **o** Expression of *CAG-EGFP* vector in the surface ectoderm (SE) after 16 h culture in vitro. Anti-EGFP (green) and TROMA-1 (magenta). **p–u** Immunohistochemistry of electroporated embryos after 24 h culture in vitro. EGFP (green), phalloidin (magenta), and DAPI (blue). Cell size enlargement and F-actin accumulation are seen in the *CAG-Grhl3* electroporated SE (**s–u**). Representative images from more than two independent experiments. Scale bars represent 10 (**p–u**), 50 (**b**, **g–l**), 100 (**c–f**), and 200 μm (**n**)

the superficial layer, i.e. SE cells of embryos. Then, electroporated embryos were cultured for 16–24 h and subjected to marker analysis (Fig. 2n–u). The plasmids, including *CAG-EGFP*, were apparently expressed in TROMA-1–positive SE cells after 16 h culture (Fig. 2n, o). In addition, these *Grhl3*-electroporated cells became multinucleated as a cluster and were enriched in F-actin (Fig. 2s–u; embryos $n = 3$; areas $n = 8$). These findings indicate that *Grhl3* appears to induce similar characters of LM-epidermal cells in vivo.

**Cytoplasmic GRHL3 transition for LM-epidermal induction.** Since the transcriptional activity of CP2 transcription factors, which include the *Grhl* family, may be regulated according to their subcellular localization[27], we hypothesized that the activity of GRHL3 in the formation of LM-epidermal cells may also similarly be controlled according to its subcellular localization. To explore this idea, we examined GRHL3 expression and found that GRHL3 protein was located in the cytoplasm of LM-epidermal cells (Fig. 3a–c, Supplementary Fig. 2). To trace GRHL3

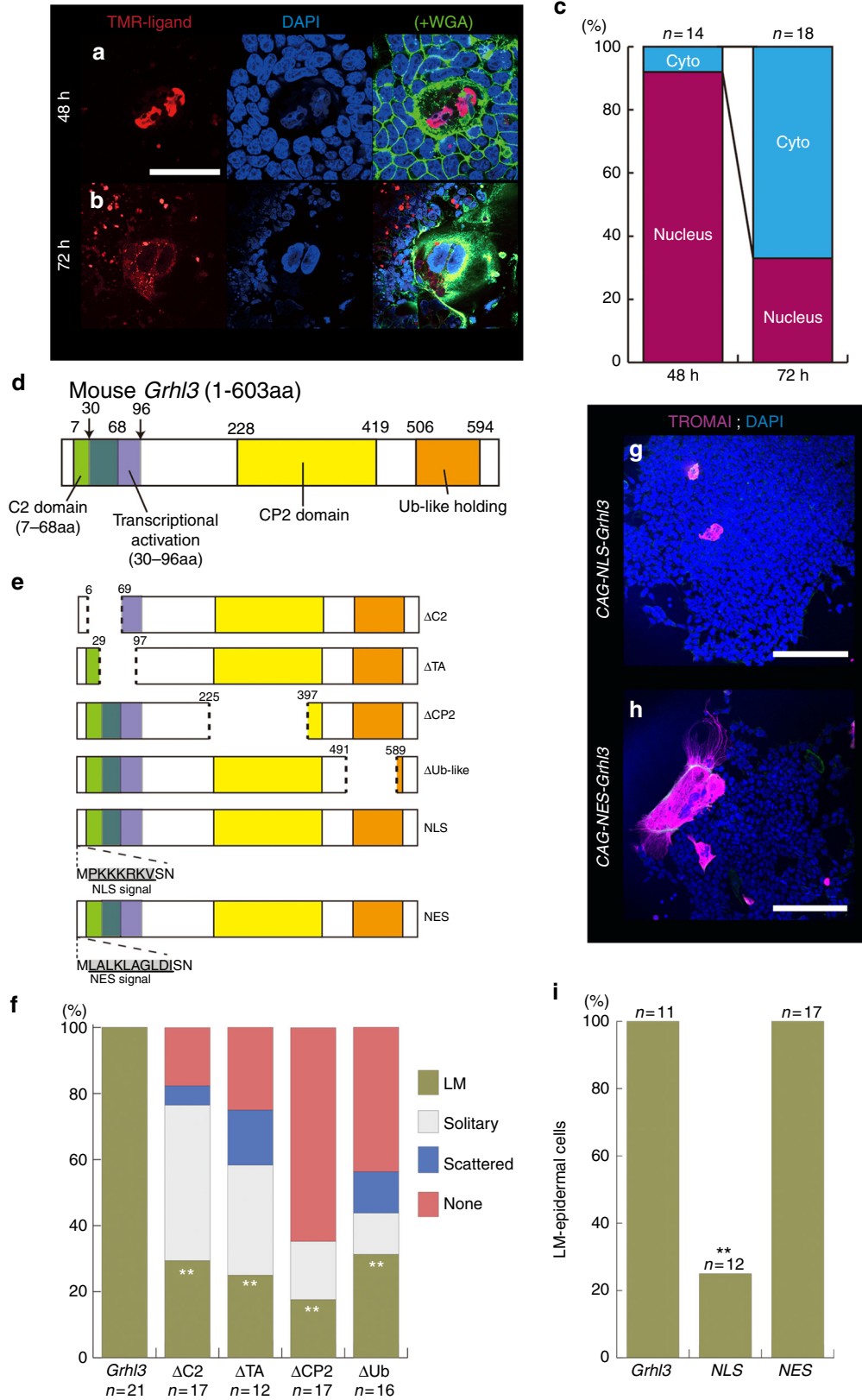

localization during epidermal maturation, a HaloTag system was designed to image protein localization in living cells (Fig. 3a–c)[28]. After a 48 h culture, HaloTag-GRHL3, as visualized by TMR-ligand, was found mainly localized within the nucleus (Fig. 3a, c, Supplementary Fig. 2). Strikingly, a further 24 h culture (72 h in

total) shifted GRHL3 expression from the nucleus to the cytoplasm (Fig. 3b, c, Supplementary Fig. 2).

To understand the mechanisms involved in the localization of GRHL3, we analyzed the subdomains directing cytoplasmic localization using a GRHL3-EGFP fusion protein with N-

**Fig. 3** Identification of the GRHL3 structure contributing to LM-epidermal formation. **a**, **b** Embryonic stem (ES) cells transfected with *CAG*-HaloTag-*GRHL3* were incubated with medium containing TMR-ligand (red). Hoechst (blue) nuclear stain. Wheat germ agglutinin (WGA, green). **c** The frequency of nuclear or cytoplasmic (cyto) localization of the GRHL3 protein observed in LM-epidermal cells using the HaloTag system after 48 h and 72 h culture. **d** Schematic illustration showing the protein domain organization of mouse GRHL3. **e** Schematic structures of GRHL3 with deletion mutations, or the insertion of NLS (nuclear import, PKKKRKV) or NES (nuclear export, LALKLAGLDI) signals are shown. **f** Summary of the induced frequency of epidermal cells after transfection of ES cells with various deletion constructs. TROMA-1–positive central epidermal cells were classified into three types: LM-, solitary, and scattered epidermal cells. **p < 0.01 ($\chi^2$ test) **g**, **h** Examples of epidermal cells induced by *CAG-NLS-Grhl3* (**g**) and *CAG-NES-Grhl3* (**h**). Immunofluorescence with TROMA-1 (magenta) merged with DAPI staining (nucleus; blue). **i** The proportions of the formation of LM-epidermal cells after transfection of ES cells with *CAG-Grhl3*, *CAG-NLS-Grhl3*, or *CAG-NES-Grhl3* plasmids. "*n*" indicates the number of EBs analyzed. **p < 0.01 ($\chi^2$ test). Representative images and frequency from more than three independent experiments. Scale bars represent 50 (**a**), and 200 μm (**g**, **h**)

terminus, middle region, and C-terminus, respectively (Fig. 3d, Supplementary Fig. 3a, b). We found GRHL3 was able to localize to the cytoplasm through the N-terminus, CP2 domain, and C-terminus, and to the nucleus through the CP2 domain and C-terminus (Supplementary Fig. 3a, b). Immunohistochemistry with specific polyclonal antibodies against mouse GRHL3 confirmed that GRHL3 localized to both the nucleus and cytoplasm in LM-epidermal cells (Supplementary Fig. 3c–j). The above findings together support the hypothesis that GRHL3 localization shifts from the nucleus to the cytoplasm during epidermal maturation (Fig. 3a–c).

To examine if the subcellular localization of GRHL3 is crucial to the induction of LM-epidermal cells, we analyzed deletion constructs of mouse *Grhl3* lacking C2, transcriptional activation (TA), and CP2 domains as well as Ub-like folding domains (Fig. 3d–f) and transfected these into ES cells. When the C2 (7–68 aa) or TA domain (30–96 aa) was deleted, induced LM-epidermal cells were significantly reduced in number compared to those induced in response to intact GRHL3 (Fig. 3d–f; 29.4% [*n* = 5/ 17], 25.0% [*n* = 3/12], respectively). These findings suggest that the N-terminus is necessary for the induction of LM-epidermal cells. Additionally, the removal of the Ub-like folding domain also reduced the emergence of LM-epidermal cells (Fig. 3d–f). Furthermore, CP2 deletion severely reduced the induction of all three types of epidermal cells: LM-, scattered, and solitary epidermal cells (Fig. 3e, f). Collectively therefore, in addition to the CP2 DNA-binding domain, N- (C2 and TA) and C-termini (Ub-like folding) that involve cytoplasmic localization are necessary for the formation of LM-epidermal cells.

To estimate whether the subcellular localization of GRHL3 itself can directly control the induction of LM-epidermal cells, we modified GRHL3 localization by the inclusion of nuclear import and export signals (nuclear localization signal [NLS] and nuclear export signal [NES], respectively) in the GRHL3 protein and then analyzed the formation of LM-epidermal cells from EBs (Fig. 3e, g–i)[29]. By adding an NLS sequence, the induction of LM-epidermal cells was markedly reduced (Fig. 3g, i). Conversely, GRHL3 carrying the NES sequence did not affect the formation of LM-epidermal cells (Fig. 3h, i). The above results together support the hypothesis that cytoplasmic GRHL3 (C-GRHL3) is crucial for the induction of LM-epidermal cells.

**Canonical and non-canonical Wnt pathways in LM-epidermal induction**. The previous finding that *Grhl3* acts as a downstream effector of Wnt/β-catenin signaling[14] led us to study whether this canonical Wnt pathway is involved in the induction of LM-epidermal cells (Figs. 1, 2, 3). To test this hypothesis, the constitutively active form of β-catenin (β-cateninS37A) was transfected into ES cells (Fig. 4a). However, β-cateninS37A failed to induce LM-epidermal cells (Fig. 4a; *n* = 0/31). Additionally, chemical modulators of canonical Wnt signaling, a Wnt agonist and an inhibitor were also employed. First, the Wnt agonist was

not able to induce LM-epidermal cells (Fig. 4b; *n* = 0/9). Next, the effects of a combination of chemical reagents and cDNA constructs (*Grhl3* cDNA) on the induction of LM-epidermal cells were analyzed. Strikingly, neither an activator nor an inhibitor of the canonical pathway (Wnt agonist and FH535, respectively) affected the emergence of LM-epidermal cells (Fig. 4e, f; *n* = 15, *n* = 17, respectively). Collectively, these findings suggest that canonical Wnt signaling is neither a downstream nor parallel pathway of *Grhl3* for the induction of LM-epidermal cells. This led us to explore additional signaling pathways that may be necessary for the induction of LM-epidermal cells.

We subsequently tested the non-canonical Wnt pathway by means of chemical modulators such as an activator (Rho/Rac/ Cdc42 activator I) and inhibitors (H-89, SP600125, and NSC23766; Fig. 4c, d, g–i). Notably, activation of the non-canonical Wnt pathway significantly induced much larger sized LM-epidermal cells compared to those induced by *Grhl3* alone (*p* < 0.05; Fig. 4c, d, g, h; *n* = 46). Moreover, inhibition of the non-canonical pathway repressed the formation of LM-epidermal cells induced by *Grhl3* (Fig. 4i). To explore the involvement of the non-canonical Wnt pathway, the dominant negative form of Dsh (*CAG-Dsh-DEP*), which specifically repressed the PCP pathway, was exploited (Fig. 4i). Consistently, the *Grhl3*-dependent formation of LM-epidermal cells was blocked by *CAG-Dsh-DEP* (Fig. 4i). Thus collectively, the non-canonical Wnt pathway appears to be involved in the induction of LM-epidermal cells as downstream or parallel pathways of *Grhl3*.

Finally, the above findings led us to hypothesize that simultaneous activation of canonical and non-canonical Wnt pathways may lead to the formation of LM-epidermal cells. To test this concept, two cDNA vectors, harboring β-cateninS37A and *Rac1*, respectively, or the constitutively active form of *Disheveled associated activator of morphogenesis 1 (Daam1)*[19,30–32], which links the PCP pathway to *PDZ-Rhogef*, were transfected simultaneously into ES cells (Fig. 4j–m). Consequently, both transfected EBs were able to form LM-epidermal cells without *Grhl3* cDNA (Fig. 4k, m; *n* = 3,8, respectively). Similarly, the simultaneous activation of both canonical and non-canonical pathways using chemicals was able to induce the formation of LM-epidermal cells (Fig. 4n; *n* = 7). These results indicate that the simultaneous activation of these two pathways was involved in the formation of LM-epidermal cells. Moreover, these findings imply that the canonical Wnt pathway facilitates the differentiation of epidermal cells while activation of the non-canonical Wnt pathway facilitates cell shape changes such as increased cell size, multi-nucleation, and the enrichment of actomyosin networks.

To explore how GRHL3 activated the non-canonical Wnt pathway, the expression of VANGL2, a PCP component, was analyzed using a HaloTag system (Fig. 5a). The expression of HaloTag-*Vangl2* was localized to the cytoplasm of an LM-epidermal cell (Fig. 5a). Moreover, considerable amounts of cytoplasmic VANGL2 apparently co-localized with GRHL3 (Fig. 5b, asterisks). To confirm the co-localization of GRHL3

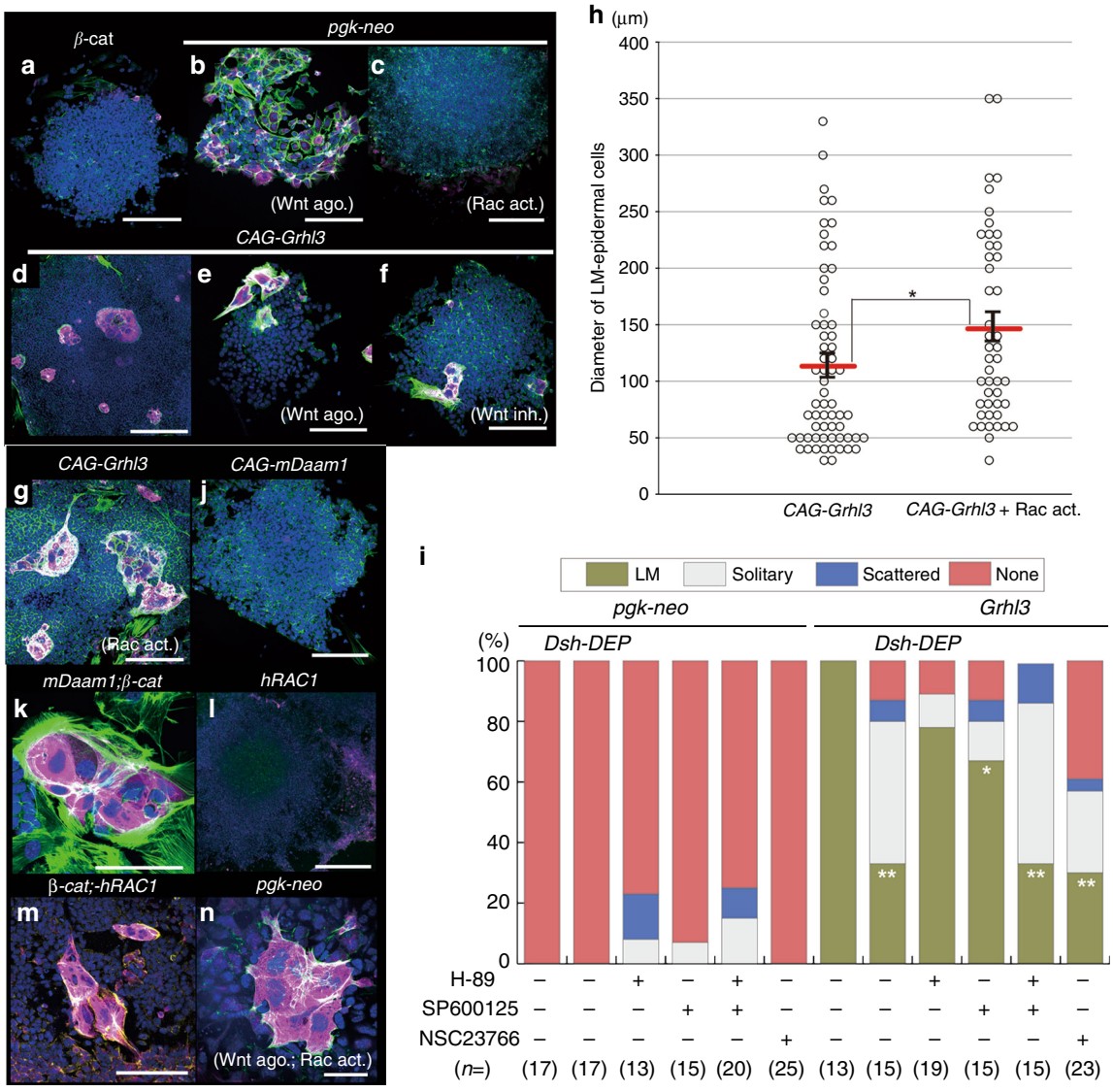

**Fig. 4** LM-epidermal induction by canonical and non-canonical Wnt signaling. **a–f, g, j–n** F-actin (phalloidin; green), TROMA-1 (magenta), and DAPI (nucleus; blue) staining of epidermal cells incubated with Wnt agonist (canonical Wnt activator; **b, e**), canonical Wnt inhibitor (FH535; **f**), Rho/Rac/cdc42 Activator I (non-canonical Wnt pathway activator; **c**, $n = 0/8$; **g**) or both (**n**). Embryonic stem (ES) cells were transfected with *CAG-β-catenin S37A* (constitutive active form) (**a**), *pgk-neo* (**b, c, n**), *CAG-Grhl3* (**d–f, g**), *CAG-mDaam1* (constitutive active form of mouse *Disheveled associated activator of morphogenesis 1 [Daam1]*)(**j**, $n = 0/21$), *CAG-mDaam1/CAG-β-catenin S37A* (**k**), *CAG-hRac1* (**l**, $n = 0/7$) or *CAG-β-catenin S37A/CAG-hRac1* (**m**). **h** Quantitative analysis of the diameters of epidermal cells induced by *CAG-Grhl3*, with or without Rho/Rac/cdc42 Activator I (146.1±12.4 μm, $n = 46$ and 113.3 ±9.8 μm, $n = 63$, respectively). Red line denotes mean ± standard error. *$p < 0.05$ (two-tailed *t*-test). **i** Summary of the induced frequency of LM-epidermal cells transfected by *pgk-neo*, *CAG-Grhl3* with/without *CAG-Dsh-DEP* and treated with the following chemical PCP inhibitors; H-89 (RhoA kinase inhibitor), SP600125 (Rac inhibitor), and NSC23766 (Rac1 inhibitor). "$n$" indicates the number of EBs analyzed. **$p < 0.01$, *$p < 0.05$ ($\chi^2$ test). Representative images and frequency from more than three independent experiments. Scale bars represent 50 (**k, m, n**), 200 (**a, b, d, e, f, g**), and 400 μm (**c, j, l**)

and VANGL2, we analyzed protein–protein interaction in NIH3T3 cells using fluorescent protein fragments of monomeric Kusabira–Green (mKG; $n = 17$; Supplementary Fig. 4). Such expression studies demonstrated that GRHL3 was located in close proximity to VANGL2 so that GRHL3 may indirectly or directly interact with non-canonical Wnt molecules in LM-epidermal cells.

In addition, expression of Scrib, another PCP protein[33], was specifically induced in the LM-epidermal cells (Fig. 5c). Coincidentally, the phosphorylation of non-muscle myosin light chain (pMLC), a marker for stress fiber formation[34], was exclusively found in LM-epidermal cells (Fig. 5d). However, although the expression of HaloTag–β-catenin was localized to both the

nucleus and cytoplasm of LM-epidermal cells, cytoplasmic β-catenin did not co-localize with GRHL3 (Fig. 5e, f). The aforementioned studies support the notion that GRHL3 may contribute to non-canonical Wnt signaling partly through expression of PCP proteins in the cytoplasm.

**C-GRHL3 is necessary for expression of PCP components**. The aforementioned features of LM-epidermal cells allowed us to investigate if such cells reflect the character of in vivo GRHL3-positive SE cells during embryonic development. To evaluate this, several molecular markers were examined by immunohistochemistry (Fig. 6). *Grhl3*-positive cells, labeled with β-

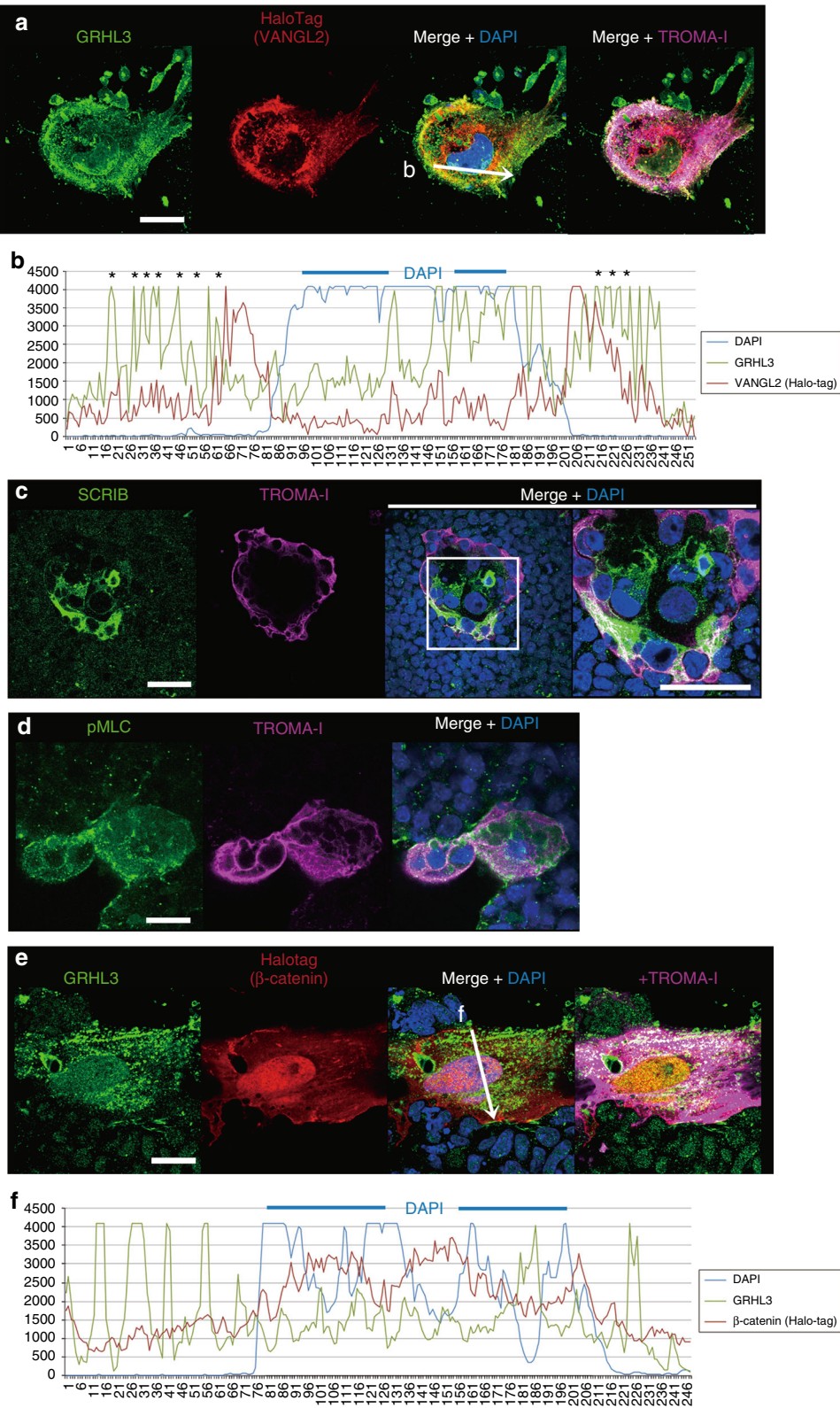

**Fig. 5** Localization of GRHL3, VANGL2, SCRIB, and β-catenin in LM-epidermal cells. **a** Immunofluorescence for GRHL3 (green), anti–HaloTag-VANGL2 (red), TROMA-1 (magenta), and DAPI (blue) in an LM-epidermal cell. **b** Intensity of merged confocal images obtained by GRHL3, VANGL2, and DAPI staining is graphically depicted (a white arrow in a). Asterisks indicate co-localization of GRHL3 and VANGL2 in the cytoplasm. **c** Immunofluorescence for SCRIB (green), TROMA-1 (magenta), and DAPI (nuclei; blue) in an LM-epidermal cell. **d** Immunofluorescence for non-muscle myosin light chain (pMLC; green), TROMA-1 (magenta), and DAPI (nuclei; blue) in an LM-epidermal cell. **e** Immunofluorescence for GRHL3 (green), anti–HaloTag–β-catenin (red), TROMA-1 (magenta), and DAPI (blue) in an LM-epidermal cell. **f** Intensity of GRHL3, β-catenin, and DAPI in merged confocal images is graphically depicted (a white arrow in **e**). Representative images from three independent experiments. Scale bars represent 50 μm

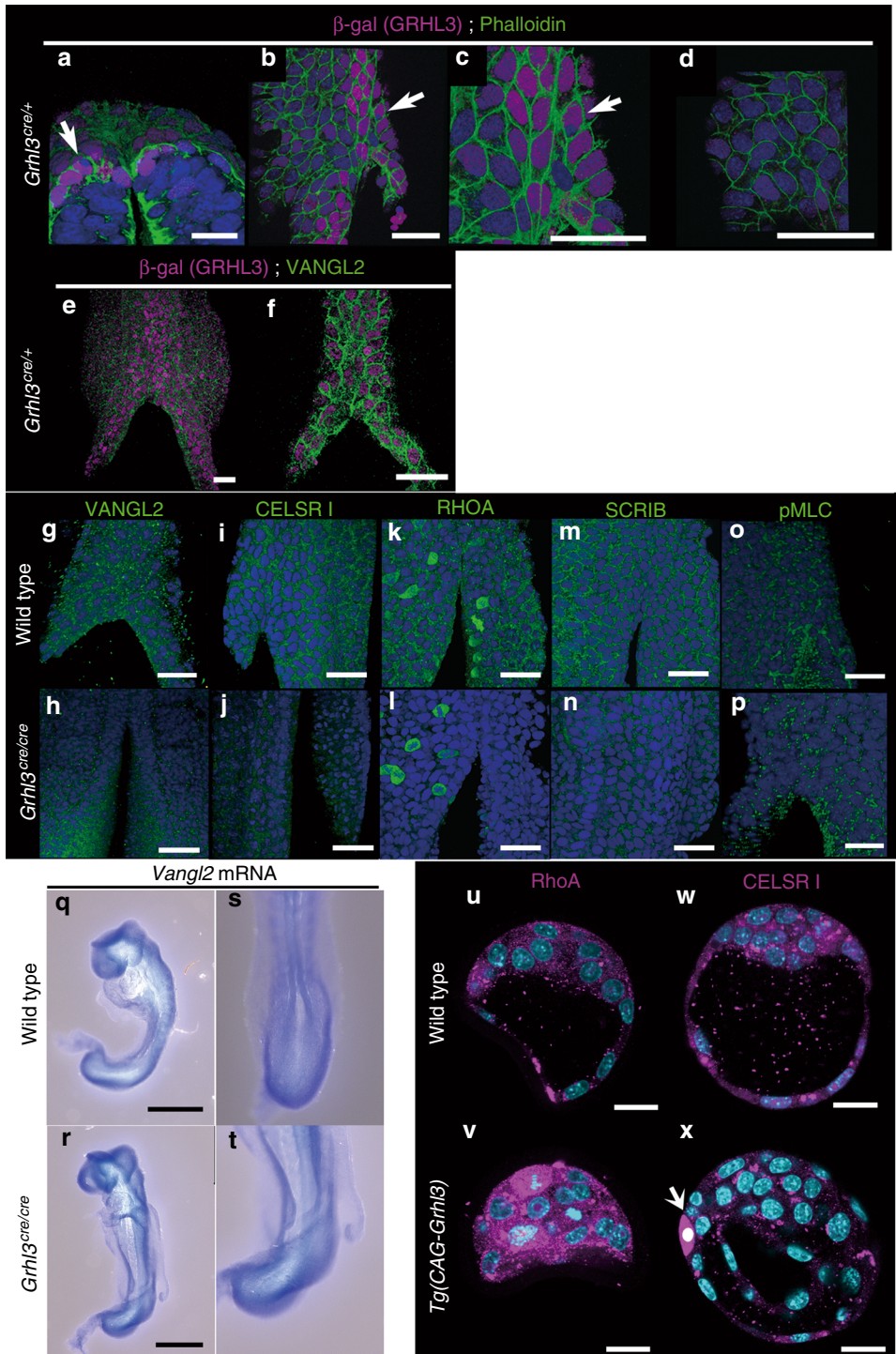

**Fig. 6** Whole-mount marker studies of mouse embryos at the neurulation stage. **a–f** Immunohistochemical and fluorescence analyses of E8.5. β-galactosidase (β-gal)-GRHL3 (magenta), F-actin (green), and DAPI (blue; **a–d**). Cre recombinase, internal ribosomal entry site (IRES), and nuclear localized β-gal were knocked in *Grhl3* in the frame of the ATG start site in the *Grhl3 cre* allele. Arrows indicate multinucleate surface ectoderm (SE) cells (**a–c**) in a *Grhl3cre/+* embryo. β-gal-GRHL3 (magenta) and VANGL2 (green) in a *Grhl3cre/+* embryo (**e, f**). **g–p** VANGL2 (green; **g, h**), CELSR1 (green; **i, j**), RHOA (green; **k, l**), SCRIB (green; **m, n**), phospho non-muscle myosin light chain (pMLC green; **o, p**) and DAPI (blue; **g–p**) in wild-type (**g, i, k, m, o**), and *Grhl3cre/cre* (**h, j, l, n, p**) embryos at E8.5. RHOA expression during cell division appeared to be unchanged in a *Grhl3cre/cre* embryo (**l**). **q–t** Whole-mount in situ hybridization of *Vangl2* mRNA in wild-type (**q, s**) and *Grhl3cre/cre* (**r, t**) embryos at E8.5. **u–x** Molecular marker analyses in *Tg(CAG-Grhl3)* embryos at E3.5. Immunohistochemical analyses of RHOA (magenta; **u, v**), CELSR1 (magenta; **w, x**), and DAPI (cyan) in wild type (**u, w**) and Tg(CAG-Grhl3) (**v, x**), respectively. Representative images from more than two independent experiments. Scale bars represent 20 (**u–x**) and 50 μm (**a–r**)

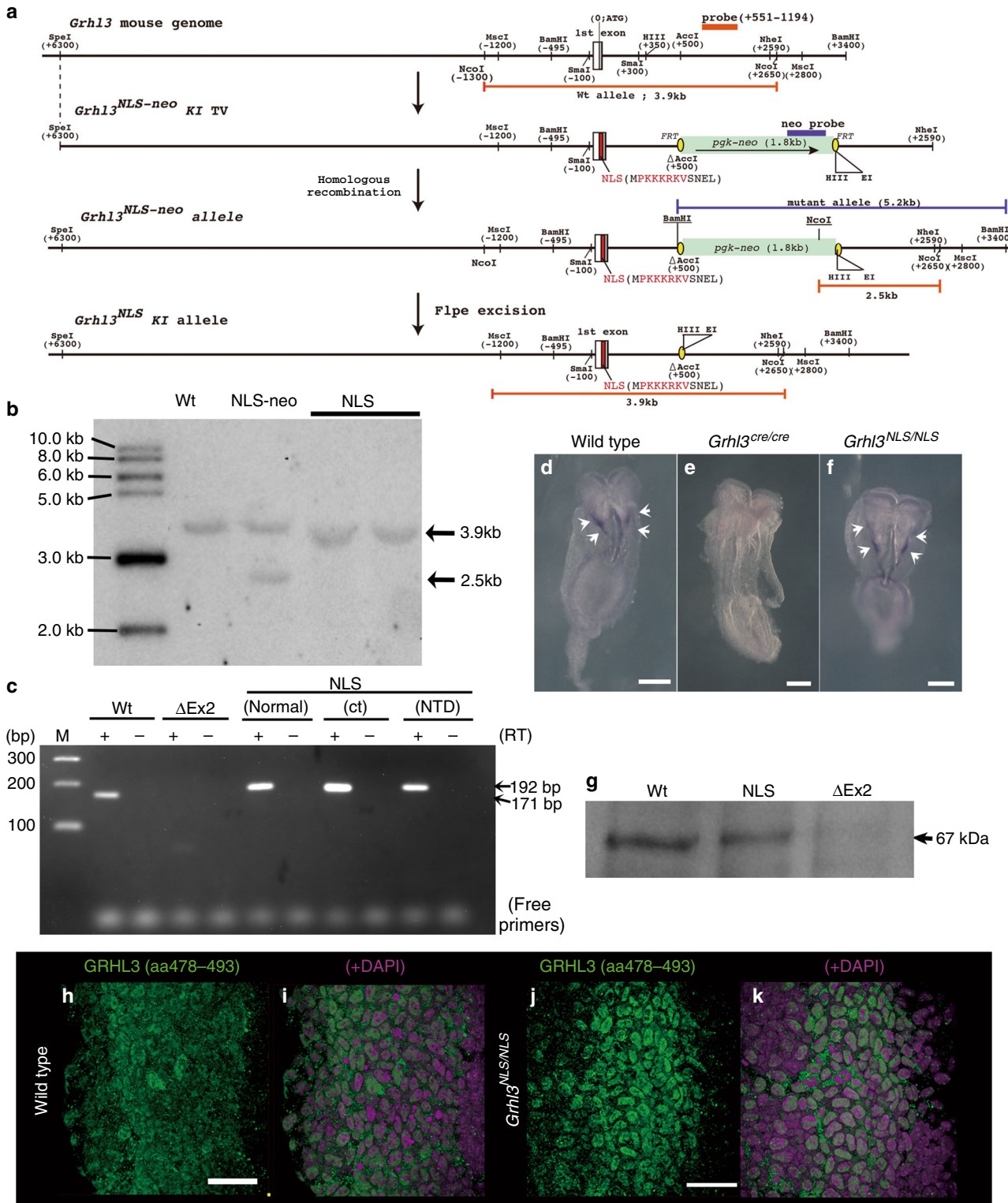

galactosidase antibody in *Grhl3cre/+* embryos, were evident in neural folds prior to neural tube closure and thereafter in dorsal SE cells[14]. In contrast, *Grhl3*-negative SE cells were distributed to the more lateral side of the epidermal layer (Fig. 6a–c)[14]. Further, F-actin and pMLC staining revealed that *Grhl3*-positive cells possessed assemblies of actomyosin bundles (Fig. 6a–d, o). Additionally, endogenous GRHL3 apparently localized to the nucleus and cytoplasm, as well as the cell membrane (Supplementary Fig. 5). Furthermore, multinuclear cells were sometimes

found in *Grhl3*-positive (β-gal positive) SE cells (Fig. 6a–c; white arrows). Such studies suggest that *Grhl3*-positive dorsal epidermal cells during neurulation may be compatible with LM-epidermal cells derived from EBs in vitro (Figs. 1–4).

Given that GRHL3 appeared to co-localize with VANGL2 in LM-epidermal cells in vitro (Fig. 5a, b), it was reasonable to postulate that *Grhl3* would genetically interact with *Vangl2* during neurulation (Fig. 6e–h). When we analyzed VANGL2 localization, we found that VANGL2 protein was detected in

**Fig. 7** Generation of an allelic series of mutations at the *Grhl3* NLS knock-in locus. **a** Strategy for the production of an allelic series of mutations at the *Grhl3* nuclear localization signal (NLS) mutant locus. A schematic representation of the *Grhl3* wild-type allele, targeting vector, *Grhl3*$^{NLS-neo}$ and *Grhl3*$^{NLS}$ alleles. **b** Identification and characterization of mutant alleles by southern blot analysis. Genomic DNA samples from wild-type (Wt) and mutant mice were digested with *Nco*I and hybridized with the DNA probe (from +551 to +1194) to examine the correct 3′ recombination and excised *pgk-neo* cassette. The 3.9-kb and 2.5-kb bands represent wild-type (Wt) and targeted *NLS-neo* alleles. The mutant 2.5-kb band shifted up to 3.9 kb after Flpe excision in the *NLS* allele. **c** The *Grhl3*$^{NLS}$ allele produced *Grhl3* transcripts from the inserted site of NLS signal. Reverse transcription (RT)–PCR analysis with cDNA from wild-type, *Grhl3*$^{ΔEx2/ΔEx2}$ and *Grhl3*$^{NLS/NLS}$ mutant embryos at E11.5. The 171-bp products correspond to wild-type transcripts, whereas the 192-bp products correspond to mutant transcripts that contain an additional *NLS* signal sequence. **d–f** Whole mount in situ hybridization analysis of *Grhl3* transcripts in wild-type (**d**), *Grhl3*$^{ΔEx2/ΔEx2}$ (**e**), and *Grhl3*$^{NLS/NLS}$ (**f**) mutant embryos at E8.0. *Grhl3* transcripts were clearly detected at the boundary between surface ectoderm and neuroectoderm in wild-type (**d**, arrows) and *Grhl3*$^{NLS/NLS}$ mutant (**f**, arrows) embryos while they were undetected in the *Grhl3*$^{cre/cre}$ embryo (**e**). **g** Western blots using the GRHL3 antibody aa195–211 (Supplementary Fig. 3g, 5c) with protein extracts of embryonic nucleus. Bands of around 67 kDa, corresponding to full-length GRHL3 products, were evident in wild-type and *NLS* alleles but not *Δex2* alleles. **h–k** Identification of nuclear-localized GRHL3 protein in *Grhl3*$^{NLS/NLS}$ mutant embryos. Immunohistochemistry of GRHL3 (anti-GRHL3 ab; aa478–aa493; Supplementary Fig. 5e; green) and DAPI (magenta). GRHL3 protein was mostly distributed in the nuclei of SE cells of a *Grhl3*$^{NLS/NLS}$ embryo (**j**, **k**), while it localized to the cytoplasm mainly and partly nucleus in SE cells of a wild-type embryo at E8.5 (**h**, **i**). Representative images from more than two independent experiments. Scale bars represent 50 (**h**, **j**) and 200 μm (**d–f**)

dorsal SE cells, specifically in the cell membrane and partly in the cytoplasm, but not in the nucleus (Fig. 6e–g). Similarly, expression of other PCP components, CELSR1, RHOA, and SCRIB, were also evident in the cell membrane of dorsal SE cells (Fig. 6i, k, m). In *Grhl3*$^{cre/cre}$ mutant embryos, however, membrane-associated VANGL2 expression was not evident (Fig. 6h) despite *Vangl2* transcripts not being dysregulated in *Grhl3*$^{cre/cre}$ embryos (Fig. 6q–t). Concordantly, the expression of other PCP components, CELSR1, RHOA and SCRIB, was reduced in *Grhl3*$^{cre/cre}$ embryos, suggesting that expression of PCP components was down-regulated by the *Grhl3* mutation (Fig. 6j, l, n). These findings suggest that GRHL3 is necessary for the correct membrane-associated localization of PCP components at the protein level. Consistent with a reduction of PCP-related molecules, pMLC expression was also down-regulated in *Grhl3*$^{cre/cre}$ embryos (Fig. 6o, p), suggesting the crucial role of *Grhl3* in the formation of actomyosin networks through PCP molecules. Concurrent with the above *Grhl3*-deficient phenotypes, expression of the PCP molecules, RHOA and CELSR1, was markedly up-regulated at E3.5 in Tg(CAG-*Grhl3*) transgenic embryos (Fig. 6u–x; *n* = 6, 2, respectively). These findings indicate that GRHL3 is necessary for the correct expression of PCP components.

To evaluate whether cytoplasmic GRHL3 was involved in epithelial morphogenesis, we generated genetically modified mice (*Grhl3*$^{NLS}$), in which GRHL3 localized to the nucleus rather than the cytoplasm, by inserting an *NLS* sequence into the translational start site of the *Grhl3* locus (Fig. 7a, b, Supplementary Fig. 6). *Grhl3* transcripts were normally generated and full-length GRHL3 was translated in *Grhl3*$^{NLS/NLS}$ embryos (Fig. 7c–g, Supplementary Fig. 7,8). Additionally, NLS-fused GRHL3 was mainly localized in the nucleus of SE cells in *Grhl3*$^{NLS/NLS}$ embryos as compared with GRHL3 in the wild type (Fig. 7h–k). Strikingly, *Grhl3*$^{NLS/NLS}$ embryos displayed neural tube defects (NTDs) at E12.5 (Fig. 8c). Moreover, differences in the degree of abnormalities were apparent between *Grhl3* null (*Grhl3*$^{Δex2/Δex2}$) and *Grhl3*$^{NLS/NLS}$ embryos (Fig. 8a–c). While spina bifida developed in all *Grhl3*-deficient embryos (Fig. 8b), the spina bifida phenotype of *Grhl3*$^{NLS/NLS}$ appeared to be milder than that of the *Grhl3*-deficient mutation; a much lower sacral spina bifida emerged in 57.1% of *Grhl3*$^{NLS/NLS}$ embryos (*n* = 12/21); a curved tail phenotype alone was present in 19%, which manifested as incomplete penetrance (Fig. 8c, o). Therefore, collectively, cytoplasmic GRHL3 is essential for correct neural tube closure.

In order to verify whether defects in *Grhl3*$^{NLS/NLS}$ embryos were a result of canonical Wnt–dependent epidermal differentiation or non-canonical Wnt–dependent epithelial morphogenesis,

we analyzed *Grhl3*$^{NLS/NLS}$ embryos more closely (Fig. 8d–n). In E9.25 wild-type embryos, ectodermal progenitor cells that did not express epidermal or neural markers were seen at the neural plate border (Fig. 8d; arrowheads)[14]. Conversely, in *Grhl3*-deficient embryos, progenitor cells were abolished (Fig. 8e)[14]. However, in *Grhl3*$^{NLS/NLS}$ embryos, such progenitor cells appeared to be present and were similar to those in wild-type embryos (Fig. 8f; arrowheads). These results indicate that epidermal specification occurred normally in *Grhl3*$^{NLS/NLS}$ embryos. Next, we found that expression of PCP molecules and pMLC were abolished in *Grhl3*$^{NLS/NLS}$ embryos as found in *Grhl3*$^{cre/cre}$ embryos (Fig. 5, Fig. 8g–n). Additionally, in agreement with the demonstration that *Grhl3* is essential for wound repair (Supplementary Fig. 9)[35], *Grhl3*$^{NLS/NLS}$ embryos did not show repair of the hind-limb amputation unlike wild-type embryos (Supplementary Fig. 9; *n* = 5). These findings suggest that defects in *Grhl3*$^{NLS/NLS}$ embryos are brought about by failure of cytoplasmic GRHL3–dependent epithelial morphogenesis involving the non-canonical Wnt pathway, but not nuclear GRHL3 (N-GRHL3)–dependent transcription involving the canonical Wnt pathway.

To investigate whether C-GRHL3 could genetically interact with *Vangl2* or *β-catenin*, we generated *Grhl3*$^{NLS}$; *Vangl2*, and *Grhl3*$^{NLS}$; *β-catenin* compound mutants and analyzed phenotypes (Fig. 8o). Remarkably, spina bifida was detected in 29.1% of *Grhl3*$^{NLS/+}$; *Vangl2*$^{Lp/+}$ double heterozygous embryos (Fig. 8o). Thus, this suggests genetic interaction between *Vangl2*$^{Lp/+}$ and *Grhl3*$^{NLS}$. Conversely, spina bifida was not evident in mice carrying *Grhl3*$^{NLS/+}$; *β-catenin*$^{+/-}$ genotypes (*n* = 0/24, Fig. 8o), although *Grhl3*$^{cre/+}$;*β-catenin*$^{+/-}$ mice exhibited a high frequency of spina bifida[14]. Taken together, these findings suggest that N-GRHL3 is defective in the *Vangl2*-dependent non-canonical Wnt pathway, but not canonical Wnt pathway.

Since *Grhl3* is known to activate the PCP pathway at the transcriptional level via *Rhogef19* expression for promoting actin polymerization[35], we verified whether failure in epithelial morphogenesis observed in *Grhl3*$^{NLS/NLS}$ embryos was due to a reduction of *Rhogef19* expression at the level of transcription or not (Fig. 8p–u). At E9.25 and E11.5, *Rhogef19* transcripts were found in the SE (Fig. 8p, s). In agreement with the previous report, *Rhogef19* transcripts were reduced in the SE of *Grhl3*-deficient embryos (Fig. 8q, t). Conversely, transcripts were not reduced in the SE of *Grhl3*$^{NLS/NLS}$ embryos (Fig. 8r, u). These additional findings support the concept that defects of epithelial morphogenesis in *Grhl3*$^{NLS/NLS}$ embryos were not brought about by down-regulation of *Rhogef19* transcripts through the transcriptional activity of N-GRHL3, but rather by the down-regulation of PCP components at the protein level through cytoplasmic GRHL3.

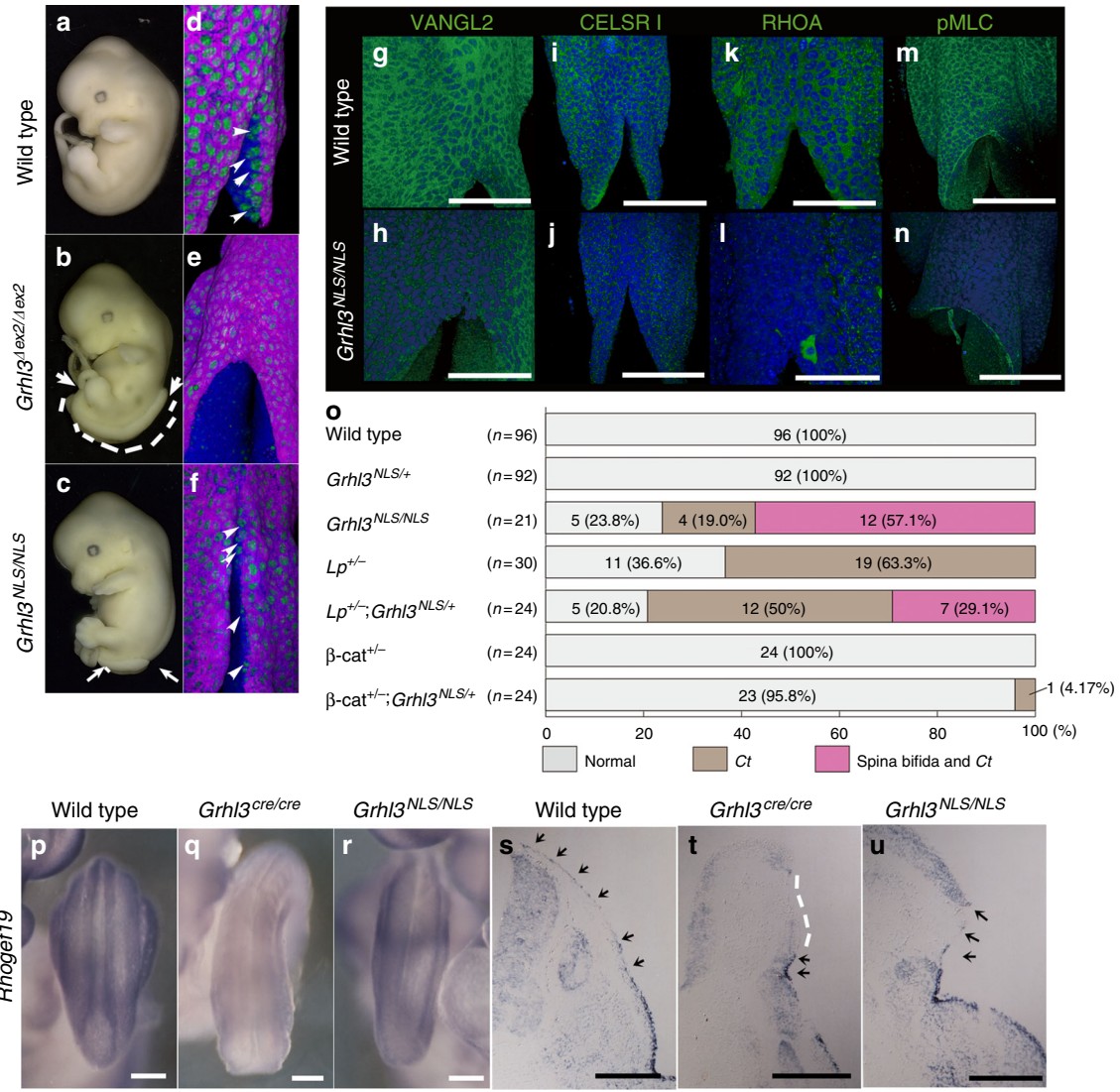

**Fig. 8** Morphological and molecular marker analyses in *Grhl3*$^{NLS/NLS}$ mutant embryos. **a–c** Gross morphology in wild-type (**a**), *Grhl3*$^{Δex2/Δex2}$ (**b**), and *Grhl3*$^{NLS/NLS}$ embryos (**c**) at E12.5. **d–n** Whole-mount immunohistochemistry of wild-type (d,g,i,k,m), *Grhl3*$^{Δex2/Δex2}$ (**e**) and *Grhl3*$^{NLS/NLS}$ embryos (**f**, **h**, **j**, **l**, **n**), at E9.25. TROMA-1 (magenta), N-cadherin (blue) and DAPI (green) in **d–f**. VANGL2 (green in **g**, **h**), CELSR1 (green; in **i**, **j**) RHOA (green in **k**, **l**), non-muscle myosin light chain (pMLC; green in **m**, **n**) and DAPI (blue in **g–n**). Arrowheads indicate ectodermal progenitor cells that did not express epidermal or neural markers at the neural plate border. **o** Variations of neural tube defect phenotypes in *Grhl3*$^{NLS/+}$ crossed with *Vangl2*$^{Lp/+}$ or *β-catenin* $^{+/-}$ mutant mice from E11.5 to E14.5. The loop-tail (*Vangl2*$^{Lp}$) allele has a strong planar cell polarity (PCP) phenotype during epithelial morphogenesis. Ct Curved tail. **p–u** Expression analysis of *Rhogef19* by whole-mount in situ hybridization in wild-type (**p**, **s**), *Grhl3*$^{cre/cre}$ (**q**, **t**) and *Grhl3*$^{NLS/NLS}$ (**r**, **u**) embryos and their transverse sections (**s–u**). *Rhogef19* transcripts were detected in the surface ectoderm (SE) region of wild-type (s, arrows) and *Grhl3*$^{NLS/NLS}$ (**u**, arrows) embryos but were absent in the surface region of the *Grhl3*$^{cre/cre}$ embryo (**t**, white dotted lines). Representative images and frequency from more than two independent experiments. Scale bars represent 100 (**k**, **l**) and 200 µm (**g–j**, **m**, **n**, **p–u**)

**GRHL3 confers mechanical properties on epidermal cells**. To explore if GRHL3-positive epidermal cells acquire mechanical cues necessary for epithelial morphogenesis, elastic properties of epidermal tissues were measured using micropipette aspiration experiments (Fig. 9a, b, Supplementary Fig. 10). Wild-type or *Grhl3* heterozygous dorsal SE tissues, in which *Grhl3* was expressed, were aspirated under 1.0 or 2.0 kPa with the same glass micropipette (100 µm diameter) for 1 min, and tissue deformation into the micropipette recorded (Fig. 9a, b, Supplementary Fig. 10). Consequently, plotted total lengths of aspirated SE aggregates indicated that the elastic property of *Grhl3*-deficient SE tissues was less stiff than that of the wild type (Fig. 9b).

Next, we measured a Young's modulus on SE tissues during neurulation using an atomic force microscope (AFM; Fig. 9c–f). We determined that the Young's modulus of SE sheets of wild-type embryos was an average of 8.328 kPa (embryos $n = 4$; areas $n = 10$), while that of *Grhl3*$^{cre/cre}$ mutant embryos was an average of 2.847 kPa (embryos $n = 2$; areas $n = 6$; Fig. 9c–f). Additionally, Young's modulus of SE cells in *Grhl3*$^{NLS/NLS}$ embryos was an average of 5.796 kPa (embryos $n = 3$; areas $n = 8$). These data indicate that SE cells expressing *Grhl3* or C-GRHL3 protein were stiffer than *Grhl3*-negative SE cells. Taken together, these results suggest that *Grhl3* is essential for ensuring mechanical properties in epidermal cells against the tensile force required for epithelial morphogenesis.

Finally, to directly evaluate the cortical tension of LM-epidermal cells induced by *Grhl3* cDNA in vitro, a Young's modulus of TROMA-1–positive epidermal cells was measured with an AFM (Fig. 9g–l)[36]. We determined that the Young's modulus of epidermal cells formed in the periphery of the EBs

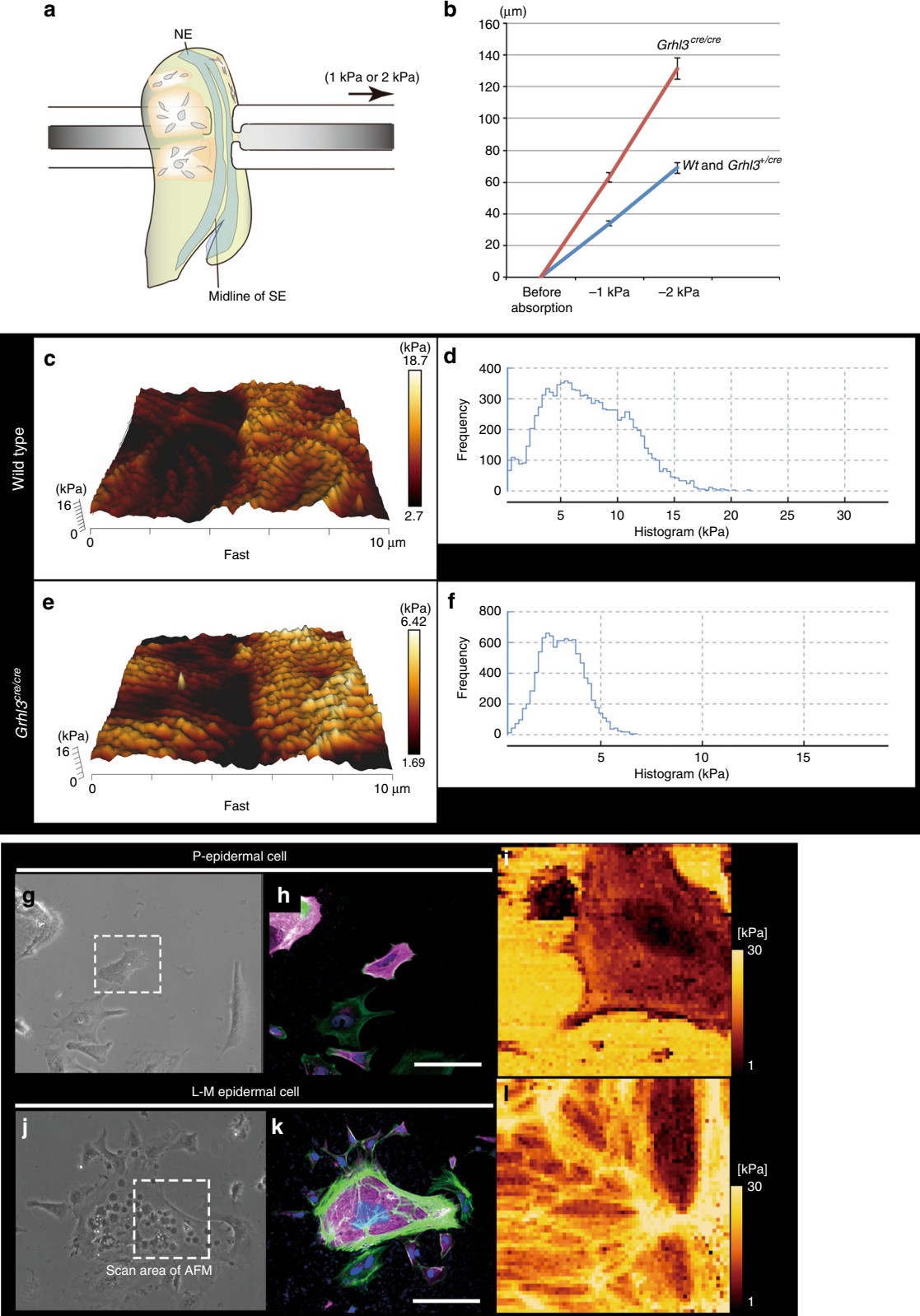

**Fig. 9** Mechanical characterization of *Grhl3*-positive and -negative epidermal cells. **a**, **b** Micropipette aspiration experiments of surface ectoderm (SE) tissues in wild-type or *Grhl3*[+/cre] and *Grhl3*[cre/cre] embryos. Quantification of the length of aspirated SE layers in wild-type or *Grhl3*[+/cre] ($n = 10$) and *Grhl3*[cre/cre] embryos ($n = 11$) from two independent experiments (**b**). Mean ± standard error. **c–f** Atomic force microscope (AFM) images (QI mode in liquid) (**c**, **e**) and stiffness profiles (**d**, **f**) of SE layers from wild-type (**c**, **d**) or *Grhl3*[cre/cre] (**e**, **f**) embryos at E9.25. **g**, **j** Bright-field microscopy images of GRHL3-negative (**g**) and GRHL3-positive epidermal cells (**j**). **h**, **k** Immunohistochemistry showing TROMA-1 (magenta), F-actin (phalloidin; green), and DAPI (blue) of cells in **g** and **j**. **i**, **l** AFM images of an epidermal cell derived from the periphery or outside of embryoid bodies (**i**, dotted square in **g**) and an LM-epidermal cell in the central region of EBs (**l**; dotted square in **j**). Representative images from more than two independent experiments. Scale bars represent 50 µm (**h**, **k**)

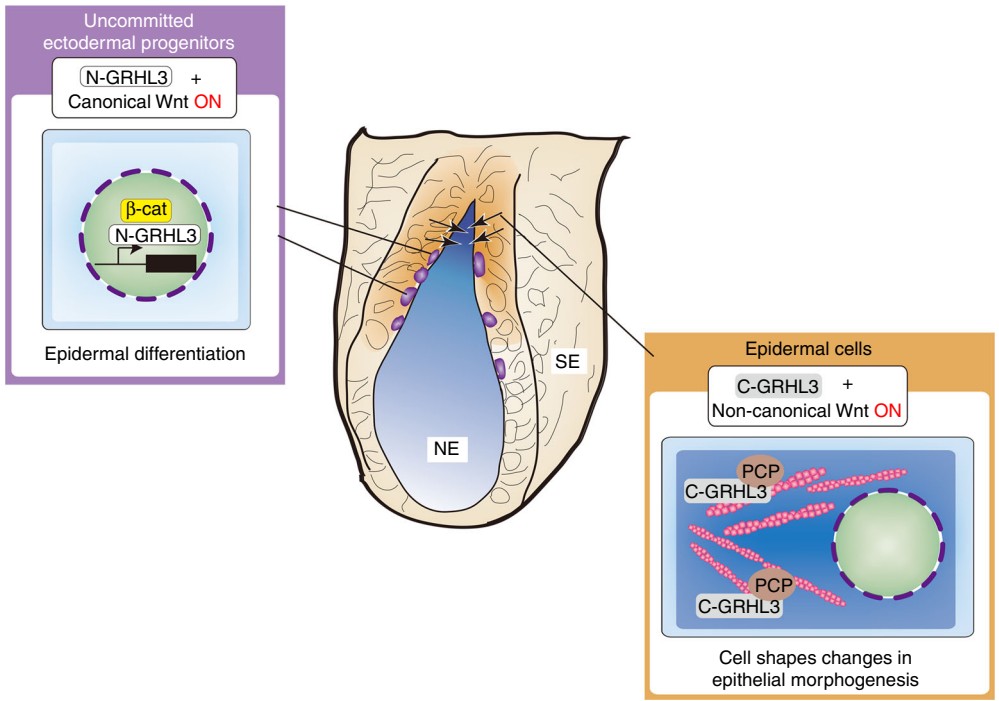

**Fig. 10** Cytoplasmic localization of GRHL3 upon epidermal differentiation triggers cell shape changes. GRHL3 exerts two different functions related to epidermal fate specification and epithelial morphogenesis during the course of epidermal development. The former function is mediated by N-GRHL3 as a downstream transcriptional factor of *β-catenin*. The latter function is mediated by C-GRHL3 involving the non-canonical Wnt pathway, e.g. planar cell polarity molecules. The non-canonical Wnt pathway facilitates actomyosin networks and, consequently, alters physical properties such as the mechanical stiffness of cells necessary for epithelial morphogenesis. The cytoplasmic localization of GRHL3 during epidermal differentiation appears to be controlled by CP2, C-terminus, and N-terminus domains of GRHL3. Thus, this cytoplasmic localization of GRHL3 triggers cell shape changes in epithelial morphogenesis from epidermal differentiation during embryonic development. SE Surface ectoderm, NE Neural ectoderm

was an average of 1.82 kPa (Fig. 9g–i; $n = 5$), while that of LM-epidermal cells was an average of 3.64 kPa (Fig. 9j–l; $n = 5$). The stiffness of LM-epidermal cells induced by *Grhl3* cDNA was therefore significantly higher than that of epidermal cells formed in the periphery ($p < 0.05$; one-tailed *t*-test). Thus, concurrent with the above in vivo study, *Grhl3* is able to confer proper physical properties on epidermal cells.

## Discussion

Our current findings indicate that N-GRHL3 directs epidermal differentiation from ectodermal progenitors and that this specification process is controlled by canonical Wnt signaling (Fig. 10)[14]. Upon epidermal differentiation, GRHL3 is localized in the cytoplasm and cell membrane, and is necessary for non-canonical Wnt-dependent cell shape changes (Fig. 10). Consequently, up-regulation of PCP molecules generates biomechanical forces primarily mediated by actomyosin networks during epithelial morphogenesis, i.e. neurulation (Fig. 10). Thus, GRHL3 functions as a unique signaling trigger for both canonical Wnt-dependent epidermal differentiation in the nucleus and non-canonical Wnt-dependent epithelial morphogenesis in the cytoplasm and cell membrane.

Cytoplasmic GRHL3 confers mechanical properties on epidermal cells for cell shape changes in coordination with cellular differentiation. Mechanical forces contribute to apical constriction and convergence toward the midline of the neural plate through actomyosin networks[37]. In this manner, mechanical forces drive neural tube closure processes such as bending and folding of the neural plate[15,38]. Therefore, it is highly probable that activation of PCP molecules directed by cytoplasmic GRHL3 facilitates actomyosin networks in epidermal cells and the

resultant mechanical properties can direct epithelial morphogenesis and remodeling for neurulation.

We have identified two separate functions for GRHL3: as a nuclear transcription factor and as a cytoplasmic transducer of cell shape changes that may be mediated by distinct protein domains, transcriptional activation, DNA-binding CP2 domains, and N- and/or C-terminal regions, respectively. The two protein functions of GRHL3 are apparently controlled during the course of embryonic development. For many years, the processes of specification and morphogenesis, crucial during development, were thought to be controlled independently by different sets of molecules. In this respect, GRHL3 may be a unique factor that plays two different roles in epidermal specification and cell shape changes at the same time through the different subcellular localization of a single protein molecule. In this manner, GRHL3 can transmit and convert those signals used in epithelial differentiation to be useful for cell shape changes during development. To date, numerous lines of evidence have revealed that specification processes are initiated by extracellular signaling, with such triggering mechanisms intensively analyzed. In contrast, molecular entities that trigger cell shape changes have not yet been clearly identified.

With respect to the polyploidy-like phenotype induced by GRHL3, it is well known that the epidermis of mammalian skin becomes polyploidy during homeostasis[39–41]. In addition, wound-induced polyploidization of the epidermis has also only recently been discovered and analyzed in the mouse;[42] epidermal cells after injury are multinucleated and become enlarged in size in order for epithelial structure to recover quickly. As *Grhl3* is also crucial for skin development as well as wound repair processes[35], it may be rational that GRHL3 contribute to multi-nucleation and the enlargement of cell size during normal development as well as wound repair.

At this stage, it remains to be determined how LM-epidermal cells induced by *Grhl3* in vitro are relevant to in vivo epidermal cells. Phenotypes of transgenic mice and electroporated cells mis-expressing *Grhl3* in vivo may not be completely identical with those of in vitro LM-epidermal cells. However, clear similarities are present between in vitro and in vivo cells. For example, multinucleation and the up-regulation of F-actin appear to be common (Fig. 2). Additionally, the expression of PCP molecules was also up-regulated both in vivo and in vitro (Figs. 5, 6). To identify in vivo counterpart of LM-epidermal cells, further comprehensive studies between LM-epidermal cells in vitro and epidermal cells in vivo during mouse development would be necessary.

This study suggests that GRHL3 may trigger both cell differentiation and shape change, a hypothesis that may provide unique insights into why cell specification is coordinated with morphogenetic processes to construct a three-dimensional complex structure during animal development. The grainy head family plays a phylogenetically conserved role in epithelial morphogenesis in the animal kingdom[16,17,43]. The capability of dual functions in the form of a nuclear transcription factor and a cytoplasmic PCP transducer of GRHL3 may also be conserved in other species. In summary, our study sheds light on some of the processes involved in the initiation of epithelial morphogenesis and its coordination with cell specification during development.

## Methods

**Epidermal differentiation from ES cells in vitro.** Epidermal cells were differentiated from G4 embryonic stem (ES) cells[44] via embryoid body (EB) cells[25]. Appearance frequency of LM-epidermal cells (%) is calculated from the number of EBs having LM-epidermal cells among total number of EBs we analyzed. The plasmids, CAG-Grhl1, CAG-Grhl3, CAG-DN-Grhl3, CAG-Grhl3ΔC2, CAG-Grhl3ΔTA, CAG-Grhl3ΔCP2, CAG-Grhl3ΔUb, CAG-Grhl3NLS, and CAG-Grhl3NES were constructed according to standard procedures (detailed procedures for plasmid constructions are available upon request). For the construction of cDNAs, the plasmids pgk-neo[45], CAG-β-cateninS37A[46], CAG-Dsh-DEP[47], CAG-mDaam1 (the carboxyl-terminal amino acids 421 to 1078 of the mouse *Daam1* cDNA; GenBank AY426535), and CAG-Rac1 (human RAC1; pFN21AB8254 from the Kasuza DNA Research Institute, Kisarazu, Japan) were used. Each cDNA, fused with a CAG promoter[48], was ligated to a neo gene driven by a pgk1 promoter with a polyadenylation signal. These plasmids were transfected into ES cells with Lipofectamine LTX (Invitrogen). Then, ES cells were cultured with G418 (150 μg mL$^{-1}$) for 24 h. The following chemical reagents were added to EB cells after transfection of plasmids: Wnt agonist (canonical Wnt activator; Merck, Kenilworth, NJ, USA; cat. no. 681665; 10 μM), FH535 (inhibitor of β-catenin/Tcf pathway; Merck cat. no 219330; 15 μM), H-89 (RhoA kinase inhibitor; CST cat.no. 371963; 10 μM), SP600125 (JNK [Rac] inhibitor; Merck cat. no. 420119, 20 μM), Rho/Rac/Cdc42 activator I (PCP pathway activator; Cytoskeleton, Inc; cat. no CN04, 1 μg mL$^{-1}$) and NSC23766 trihydrochoride (RacI inhibitor; Funakoshi cat.no. A12546-10 100 μM).

**HaloTag labeling.** HaloTag labeling was performed according to the manufacturer's directions (www.promega.co.jp). We obtained Flexi HaloTag clones, pFN21AE2519 (human GRHL3), pFN21AB6277 (human β-CATENIN), and pFN21ASDA1215 (human VANGL2) from the Kazusa DNA Research Institute and halotag TMR ligand (promega cat.no. G8252) as halotag ligand.

**Cell lines.** G4 ES cells were obtained from Dr. Andras Nagy (Lunenfeld-Tanenbaum Research Institute at Mount Sinai Hospital in Toronto) and NIH3T3 cells were obtained from the Japanese Collection of Research Bioresources (JCRB, Japan).

**Immunohistochemistry and in situ hybridization.** Information of primary antibodies used in this study is described in Supplmentary Table. Specifically, regarding the antibody against mouse GRHL3, since the aa478-493 antibody displayed less background staining and recognized cytoplasmic GRHL3, the aa478-493 antibody is suitable to detect the translocation of GRHL3 from the cytoplasm into nucleus, i.e. it is appropriate to examine how much cytoplasmic GRHL3 is reduced or not. Consequently, the aa478-493 antibody was used to analyze the distribution of GRHL3 in *Grhl3*$^{NLS/NLS}$ mutant embryos. In situ hybridization involving digoxigenin-labeled probes was conducted in a manner identical to that of Wilkinson[49].

**Generation of a nuclear localization signal knock-in mouse.** All constructs used this study were generated by standard molecular cloning techniques. In brief, a fusion gene consisting of the 5′ noncoding region of the nuclear localization signal (NLS)[29], the translational start site of the *Grhl3* gene, and a *neo* cassette flanked with *loxP* was constructed (detailed procedures for this construction are available upon request). In the targeting vector, the lengths of the homologous region were 6.3 and 2.5 kb at the 5′ and 3′ sides of the insert, respectively. Homologous recombinant G4 ES cells and chimeric mice were generated by microinjection. To delete the *loxP*-flanked *neo* cassette, male chimeras were mated with C57Bl/6 background females carrying a *Flpe* gene driven by the chicken β-actin promoter[50].

**Generation of CAG-Grhl3 transgenic mice.** Mouse *Grhl3* cDNA fused to the CAG promoter was ligated to the *LacZ* cassette flanked by two *loxP* sequences[51]. The resultant construct, CAG-loxP-lacZ-loxP-Grhl3 cDNA (5 ng μl$^{-1}$) was micro-injected into the pro-nucleus of the fertilized eggs from superovulated CD-1 female mice and the injected eggs were cultivated in KSOM overnight then transferred to the pseudopregnant CD-1[44]. The resultant offsping are genotyped to identify Tg (CAG-lacZ-Grhl3) founders. Tg(CAG-Grhl3) embryos were obtained by crossing Tg (CAG-lacZ-Grhl3) mice with β-actin Cre mice[14].

**Mouse genotyping.** *Grhl3* NLS knock-in founders and their progenitors were routinely determined by PCR and confirmed, when necessary, by southern blots of genomic DNA samples prepared from mouse tails or yolk sacs. In PCR analyses, primers and lengths of products were as follows: knock-in *NLS* sequence in the *Grhl3* allele *Grhl3-cre* (5′-AATTAAGAGACGAGTGGTCAGCAGCAGCGCCTG-3′) and *Grhl3-cre wt rev* (5′-ACCCTTACAAATTGCCGTGTGAATCCGGGC-3′), yielding 213 bp as the wild-type allele and 234 bp as the *NLS* knock-in allele.

*Grhl3-Cre-IRES-nlsLacZ* knock-in mice were obtained from the Mutant Mouse Regional Resource Center (MMRRC) and genotyped with the following primers: *Grhl3-cre* (5′-AATTAAGAGACGAGTGGTCAGCAGCGCCTG-3′), *Grhl3-cre wt rev* (5′-ACCCTTACAAATTGCCGTGTGAATCCGGGC-3′), and *Grhl3-cre mut* (5′-GCAGCCCGGACCGACGATGAAGCATGTTTA-3′), yielding 213 bp as the wild-type allele and 370 bp as the mutant allele[52].

*Grhl3*$^{ΔEx2}$ mice were obtained by crossing *Grhl3*$^{tm1a(EUCOMM)Wtsi}$ stock (International Mouse Phenotyping Consortium) with Cre-expressing mice[53]. In PCR analyses, the primers and lengths of products were as follows: flox-Ex2-A2 (5′-TTTGGGTGATGATGGGATCAGACCCAGGTC-3′), flox-Ex2-B2 (5′-CCATTCAAGCAATGTGTGCTGTACACCAGG-3′) and flox-Ex-D2 (5′-CCCCGGATCTAAGCTCTAGATAAGTAATGA-3), yielding around 1050 bp as the wild –type allele and 300 bp as the *Grhl3*$^{ΔEx2}$alelle.

β-catenin mutant mice were obtained from the Jackson Laboratory (Bar Harbor, Maine, USA)[54]. An β-catenin mutant allele was identified as follows: RM41-internal (5′-AAGTTGTTTGTACAGAGTGTGGAGTTACTA-3′), RM42-internal (5′-CTCTCTGCCCAAGTGTAAACTTATGAGGCC-3′), and RM43-internal (5′-GGTATGTACAACATTGTTGGAACTTAGACA-3′) yielding around 200 bp for the wild-type allele and around 500 bp for the mutarant allele.

An *Lp* mutant allele (*Vangl2*$^{Lp-2}$) obtained from the Jackson Laboratory) was identified as follows: Vangl2Alu-for1 (5′-CAACAGTATCTTCTCCCTTCCTCAGGCCT-3′) and Hpy166IIrev1: (5′-TCCTCAGAGAGTTTGAAGAAGGGCACCTTC-3′), yielding around 230 bp for the wild-type allele and the *Lp* mutant allele[55]. Next, the 230 bp DNA fragments were purified with an illustra$^{TM}$ GFX$^{TM}$ PCR DNA and Gel Band Purification kit (GE Healthcare; cat. no 28903470). Subsequently, the purified fragments were digested with *Hyp*166II restriction enzyme and classified as wild type (125 bp and 100 bp), *Lp* heterozygous (125 bp, 115 bp, and 100 bp) or *Lp* homozygous (115 bp and 100 bp) after electrophoresis in UltraPure Agarose-1000 (ThermoFisher; scientific cat. no 16550-100).

Transgenic mice, CAG-lacZ-Grhl3 and CAG-Grhl3, were routinely determined by PCR. PCR primers and lengths of the products in the PCR analyses were as follows; in the transgenic mice, CAG-lacZ-Grhl3 was identified with primers CAG-Pro (5′-TAGAGCCTCTGCAACCTGTTCATGCCTT-3′) and CAG-lacZ (5′-AGTGTCCCAGCCTGTTTATCTACGGCTTAA-3′), yielding 270 bp. The CAG-Grhl3 allele excised by Cre protein was determined with primers CAG-Pro2 (5′-TACAGCTCCTGGGCAACGTGCTGGTTGTTG-3′) and CAG-Grhl3 (5′-CTTTCCTTGGTCATTCCGGCCACCAGTGC-3′), yielding 300 bp.

**Whole embryo electropolation ex utero.** *Ex utero* whole–embryo (E8.75) electroporation was conducted as follows[56]. In brief, dissected tissues of ICR strains of mice were soaked in culture medium (50% rat serum and 50% Tyrode's saline buffer). To prepare embryos for electroporation, decidua was trimmed away and two types of DNA solution, CAG-EGFP (1 μg mL$^{-1}$), with or without CAG-Grhl3 (1 μg mL$^{-1}$), were injected into the amniotic cavity using a glass capillary. Electroporation was conducted at a voltage of 70 (5 cycles of 50 msec pulse length and 950 msec interval) in Tyrode saline buffer using a CUY21EDITIII electroporater and LF650P3 electrode (BEX, Tokyo, Japan). After electroporation, embryos were cultured in the above culture medium at 37 °C in a 5% CO$_2$ incubator.

**Histology**. For histological analysis, embryos and EB cells were fixed in Bouin's fixative, dehydrated, and embedded in paraplast. Serial sections were generated and stained with hematoxylin and eosin.

**Western blot analysis**. Nuclear proteins from SE tissues of E16.5 were extracted using a Subcellular Protein Fraction Kit (ThermoFisher, cat. No.78840). Nucleo-protein extracts were quantified using a Qubit 3 Fluorometer (ThermoFisher), after which 40 µg of each extract was loaded onto a 10% bis-acrylamide gel and subjected to SDS-polyacrylamide gel electrophoresis (PAGE). Proteins were transferred to a polyvinylidene fluoride membrane using a Trans-Blot turbo transfer system (BioRad). The membrane was blocked using Block Ace Powder (DS Pharma Biomedical) for 2 h at RT. After primary incubation using blocking solution with anti-GRHL3 antibody (1:1000 antigen aa195-211) overnight at 4 °C, a second incubation was performed using blocking solution with anti-rabbit HRP antibody (Promega). Finally, the signal was revealed using a Western Lightning ECL Pro Kit (Perkin Elmer, Waltham, MA, USA; cat. No. NEL120001EA).

**Reverse transcription (RT)-PCR analysis**. Total RNA was isolated from wild-type, $Grhl3^{\Delta Ex2/\Delta Ex2}$ and $Grhl3^{NLS/NLS}$ mutant embryos at E11.5 with TRIzol reagent (Thermo Fisher cat.no. 15596026), respectively. First-strand cDNA synthesis was performed with oligo-dT primers and Superscript III First-Strand synthesis System for RT-PCR (Invitrogen cat.no. 18080-51). The cDNA was utilized as a PCR substrate by standard protocols. The PCR primers, the forward primer (5′-ACCAGAGACGGATCGCTGGAACCTCGGAGA-3′) and the reverse primer (5′-CTGGAGAACCCTTTGACGGCTGCCACCAAA-3′) were employed for amplification.

**Atomic force microscopy**. Young's modulus of epidermal cells in vitro was measured using atomic force microscopy (AFM). AFM and the following statistical analysis have been described previously[57,58]. Epidermal cells differentiated from ES cells via EBs (embryoid bodies) were transfected with a $Grhl3$ plasmid and cultured on a glass substrate. AFM was used to measure Young's modulus of epidermal cells. After the AFM measurement, cells were fixed and stained with TROMA-I–specific antibody. TROMA-I–positive cells were considered epidermal cells and used for statistical analysis. Cells with an area smaller than 3000 µm² were considered peripheral epidermal cells, and cells with an area larger than 10,000 µm² were considered large and mature (LM)-epidermal cells.

Young's modulus of mouse embryonic SE cells was measured by AFM using a NanoWizard IV imaging system (JPK Instruments) combined with an inverted optical microscope (Olympus; IX73). AFM was operated in the QI mode in liquid by adjusting the contact force during imaging with a SQUBE type cantilever (type CP-cont-BSG-B). Post-processing was conducted using JPK Image Processing; in any case, the post-processing was minimized so to reduce processing artifacts.

**Micropipette aspiration experiments**. In the micropipette aspiration assay, two opposing glass micropipettes (about 100 µm diameter) were connected with a pressure control system through a manipulator (Narishige; cat. no. MMO-202ND and IM-11-2) and held against the dorso-ventral region of each embryo. Aspirations, (1 kPa and an additional 1 kPa to total 2 kPa) measured by digital pressure gauge (Krone, KDM30), were applied to the dorsal region of embryos to stably hold SE tissues. After 1 min aspiration, the height of absorbed SE tissues was recorded with an Olympus FluoView™ FV1000 and × 40 zoom, and calculated by Olympus software.

**Embryonic wound repair assays**. Analysis of wound healing repair was performed on E12.5 wild type, $Grhl3^{cre/cre}$ and $Grhl3^{NLS/NLS}$ mutant embryos[59]. After amputation of hind-limb, wound repairs were judged by the repaired epidermal region covered on the original wound diameter marked by TROMA-I expression with immunohistochemistry and scanning electron microscope (SEM) (Miniscope TM3030Plus Hitachi).

**Animal experiments**. All mouse studies followed the fundamental ethical guidelines for proper conduct of animal experiments and related activities in academic research institutions under the jurisdiction of the Ministry of Education, Culture, Sports, Science, and Technology of Japan and were approved by institutional committees at the Research Institute, Osaka Women's and Children's Hospital for animal and recombinant DNA experiments.

**Statistical analysis**. We analyzed cell-based assays and animal data by Student's $t$-test or $\chi^2$ test and defined statistically significant differences as $p < 0.05$.

## Data availability

All relevant data are available from the corresponding authors upon reasonable request.

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

## Acknowledgements

We are grateful to Drs. M. Tada, A.M. Zorn for the plasmids, Dr. A. Nagy for G4 ES cells, Dr. Y. Ueda for the technical support of AFM and Dr. M. Montcouquiol for the polyclonal VANGL2 antibody. The monoclonal antibody TROMA-I, developed by P. Brulet, and R. Kemler was obtained from the Developmental Studies Hybridoma Bank, created by the NICHD of the NIH and maintained at The University of Iowa, Department of Biology, Iowa City, IA 52242. The *CAG-FLPe* mouse strain (RBRC01834) was provided by RIKEN BRC through the National Bio-Resource Project of the MEXT, Japan. This work was supported in part by a grant-in-aid for Scientific Research on Priority Areas and on Innovative Areas (JP16H01456, JP18H04771, and JP17H05783) and Scientific Research (B)(JP26291053) and Challenging Exploratory Research (JP15K15408) from the Ministry of Education, Culture, Sports, Science, and Technology, Japan; and by the Kato Memorial Bioscience Foundation and the Takeda Science Foundation. The funders had no role in the study design, data collection and analysis, decision to publish, or preparation of the manuscript.

## Author contributions

C.K-Y. and I.M. initiated the research and planned the experiments. C.K-Y. and K.M. performed most of the experiments. T.M. performed AFM measurement of epidermal cells. M.N. performed construction and characterization of the mouse *Daam1* plasmids. C.K-Y. and I.M. contributed to the writing of the manuscript.

## Additional information

**Competing interests:** The authors declare no competing interests.

