## [Peer Review File · Nature Communications]

Reviewers' comments:

Reviewer #1 (Remarks to the Author):

The manuscript by Yoshida et.al proposes a new mechanism by which the nuclear transcription factor Grhl3 transitions from the nucleus to the cytoplasm during embryogenesis triggering epithelial morphogenesis.

The paper is potentially significant because it proposes a non-transcriptional mechanism by which Grhl3 is required for planar cell polarity during embryogenesis. Grhl3 has been previously linked to planar cell polarity but through transcriptional mechanisms. Therefore, this study proposes a new, radically different mechanism. The authors present experiments with mice (Grhl3^{nsl/nsl}) where they knocked in an extra nuclear localization signal into the Grhl3 transcript and showed that a phenotypic abnormality (defect in neural tube closure) that is similar, however milder, than what was observed in Grhl3 deleted mice Grhl3^{-/-}. Presumably, Grhl3 does not localize to the cytoplasm in this mouse. This supports the proposed model where cytoplasmic translocation of Grhl3 during embryogenesis is required.

The weakness of the paper is that most of the figures deal with an in vitro model where overexpression of Grhl3 leads to the formation of big, multinucleated cells, apparently in response to cytoplasmic localized Grhl3. The problem is that these cells seem highly artificial and the authors never show such cells in vivo. This is an interesting finding but because of it is an in vitro cell culture phenomenon, it is hardly of much general interest. The mouse experiments are stronger but they also have some weakness as discussed below.

Another issue with this paper is the writing. It is difficult to understand how the authors use the term morphogenesis. Conventionally, this term is used to describe the formation of an organism, an organ or a part of an organ. As such morphogenesis usually depends on multiple cellular features such as proliferation, migration, differentiation, etc. Therefore, it is strange to state "triggers epithelial morphogenesis from differentiation". It appears that the authors use the term to describe cell shape changes, which is not traditional. Also, since Grhl3 has a role in the terminal differentiation of the interfollicular epidermis during embryogenesis, the authors might make it more clear upfront (at least in the abstract) that when they are talking about epidermal differentiation, they are talking about the early commitment to the surface epidermis.

Other comments:

1. In Figure 1, the authors noticed the large mature epidermal cells induced by Grhl3 cytoplasmic translocation which don't seem to have biological relevance in live animals during development. The authors should apply experimental controls to eliminate artifacts that are induced by in-vitro transfection (over-expression of Grhl3). In addition, overexpressing Grhl2, doesn't mean is endogenously expressed in EB and p-epidermal cells (as stated by the authors in the text).
2. In Figure 2, the authors should include experimental statistics (number of biological replicates showing halotag-grhl3 cytoplasmic translocation, number of biological replicates showing reduction in LM epidermal size in CAG-NLS-Grhl3 transfected cells).
3. In Figure 3, the authors claim that activation of PCP is necessary for LM-epidermal cells formation and inhibiting PCP blocks Grhl3-dependent LM epidermal cells formation. The authors don't show clear PCP activation exclusively in LM-epidermal cells compared to other cells in the EB (expression of PCP markers). When PCP was inhibited (CAG-Dsh-DEP), LM epidermal cells were still formed (~ 70% of total EBs), this data dispute the authors claim. In addition, expression of Grhl3 and Vangl2 in LM epidermal cells cytoplasm is not enough data to conclude that grhl3 contributes to PCP via vangl2.
4. In Figure 4, the authors claim that Grhl3 deletion led to abolished Vangl2 protein expression, however, the provided IF images show normal staining of Vangl2 in Grhl3^{-/-} embryos. Also, the authors need to be consistent in IF images magnifications (Figure 4 a-e). It is surprising that staining with Grhl3 antibody show no fluorescence background in Grhl3^{-/-} embryos. The authors must include negative control (no antibody) staining WT embryos to validate Grhl3 antibody staining.
5. In Figure 5, the authors have generated a mouse model where Grhl3 expression is restricted

only to the nucleus. The authors have to show that Grhl3 is normally expressed in these embryos and that Grhl3 is restricted to the nucleus by staining for Grhl3 in Grhl3^{NSL/NSL}, Grhl3^{-/-} and wild type embryos where Grhl3 is expected to be cytoplasmic like claimed. The authors have to elucidate that the phenotype observed in (Grhl3^{NSL/NSL}) is not due to alteration in the ability of Grhl3 to transcriptionally regulate genes involved in neural tube closure and epidermal differentiation. The authors claim that Grhl3 is necessary for correct membrane localization of VANGL2 at the protein levels based on IF staining in vivo, did the authors observe similar phenotype in vitro if Grhl3 is mutated? In addition, how did the authors determine that wound healing was delayed in Grhl3^{NSL/NSL} embryos? The author should state the parameters of wound healing measurements and include a figure with quantifications and biological replicates.

6. In Figure 6 the authors claim that cytoplasmic localization of Grhl3 is required for PCP and mechanical strength, the authors should include Grhl3-NSL embryos in the experiment to support the proposed hypothesis. In addition, the technique using whole embryo to measure the mechanical strength doesn't exclude the involvement of other non-epithelial cells in the data collection. The authors must use appropriate controls for the experiment.

7. Lastly, Grhl3 is known to regulate PCP on the transcriptional level by directly binding to RhoGEF19 promoting actin polymerization, the authors need to show that this is not the case here and that Grhl3^{NSL/NSL} doesn't alter the expression of RhoGEF19 or polymerization of actin.

Reviewer #2 (Remarks to the Author):

In this manuscript Kimura-Yoshida et al explore the role of GRHL3 in epidermal specification. The authors find that in addition to its previously reported role in b-catenin mediated specification of epidermal fate, GRHL3 is also important for epidermal maturation via modulation of PCP signalling. Most notably the authors show in vivo that the consequence of these interactions is a change in the biomechanical properties of the epithelia.

This is a very interesting manuscript that brings new light into how the epidermis differentiates. The manuscript would however greatly benefit from better quantification of a number of assays and a clearer description of some of the data. Some of these issues are detailed below:

The introduction lacks a lot of background detail regarding epidermal cells specification and differentiation. Also, some background on what is different between LM- and P-epidermal cells would be good.

Figure 1f – The authors argue that a dominant-negative form of Grhl3 cDNA (DN-Grhl3), reduces the number of LM-epidermal cells (Fig. 1e,f), but in both control transfected and DN-Grhl3 transfected EBs this number appears to be close to 0, therefore it is not possible to conclude that the DN inhibits LM-epidermal differentiation.

Figure 2a-b- The authors need to provide quantification of what proportion of cells have nuclear and cytoplasmic GRHL3 localization at 48 and 72 hours post-transfection.

The labelling of some of the figures is not clear, for example in Figure 3a-h, it takes a while to work out which treatment the cells received

Extended Figure 3b. The authors need to quantify the ability of beta-catenin to induce LM-epidermal cells.

Figure 3 and page 9 of the text appear to contradict each other. The text suggests that PCP activity is modulated independently of Grhl3, but the labelling of the figure suggests otherwise. The authors need to clarify with proper quantification which are the effects that manipulating PCP alone has on LM-epidermal cell differentiation and which effects does manipulating PCP and over-expressing Grhl3 have on this differentiation. Similarly the effects that b-catenin and PCP manipulation alone and together need to be properly quantified.

Figure 3a-b The authors need to quantify the effects of PCP modulation alone on LM-induction.

Figure 3k-m. The authors need to provide quantification for the different co-localization experiments.

Reviewer #3 (Remarks to the Author):

The manuscript by Kimura-Yoshida addresses the role of Grhl3 in epithelial morphogenesis by combining *in vitro* and *in vivo* approaches. *In vitro*, the authors show that forced Grhl3 expression can induce large epidermal cells that are enriched in actomyosin. Induction of these large epidermal cells appears to require cytoplasmic Grhl3 and can be blocked by a dominant-negative version of Dsh. *In vivo*, a mutant form of Grhl3 that is fused to NLS interacts with Vangl in double heterozygous animals, resulting in spina bifida in about a third of the embryos. Double heterozygous of Grhl3-NLS and beta-catenin did not show this phenotype. Finally, the authors use micropipette aspiration assays and AFM to measure and compare mechanical properties in Grhl3 expressing and non-expressing cells *in vivo* and *in vitro*. The authors conclude that the cytoplasmic pool of Grhl3 activates the PCP pathway to allow epidermal cells to acquire mechanical properties important for morphogenesis.

The manuscript addresses an interesting and important question with regard to the relationship between cell differentiation and morphogenesis. The strength of the manuscript is the use of different mouse mutants to test for genetic interactions. However, as outlined in detail below, a number of claims by the authors are not well supported. The authors central claim that cytoplasmic Grhl3 controls morphogenesis, for example, and the link between Grhl3 and PCP are weak. Moreover, there is little mechanistic understanding of how cytoplasmic Grhl3 could control morphogenesis. The conceptual advance of this work is therefore somewhat limited.

Specific comments

Fig. 1b-e. Control and experiment are difficult to compare. The authors should show images at the same magnification and stained for the same markers. It is unclear how the authors distinguish between P-epi and LM-epi. Simply by size?

What is the transfection efficiency? Do all cells express Grhl3 or only those cells that differentiate into LM cells? The authors should include a marker for transfected cells.

What is the *in vivo* counterpart of LM-cells?

Fig. 1f. I am puzzled that Grhl3 induces 100% LM cells. In Fig1c, it appears that only a small fraction of EB cells differentiated into LM cells. The authors need to clarify.

Fig. 1e,f. The authors claim that DN-Grhl3 reduces the number of P-epidermal cells and refer to Fig. 1f. However, Fig1f details the number of LM, but not P epidermal cells. The authors should show a separate quantification of P epidermal cells.

Fig. 1b-b", d. The authors claim that GRHL2 is endogeneously expressed in EBs as well as P-epidermal cells and refer to Fig. 1b-b",d. However, these Figs do not show GRHL2 staining. The authors' conclusion at the end of this paragraph that both Grhl2 and Grhl3 are able to contribute to epidermal contribution are not supported by data.

Page 6. "Since the transcriptional activity of CP2 transcription factors..." How is this related to Grhl3?

Fig. 2/ Page 6 The authors should stress that GRHL3 expression is analysed in GRHL3 transfected cells.

Page 7, top. The authors should mention that the analysis of the localization of GRHL-3 subdomains was done in MCF7 cells, a breast cancer cell line. Could the authors transfect these GFP-tagged subdomains into ES cells and then analyse subcellular localization in epidermal cells?

Extended Fig. 2. The authors claim that GRHL3 localized to the cytoplasm through N and C termini, and to the nucleus through the middle region. However, this Fig. shows that the C-terminus directs both cytoplasmic and nuclear localization. The middle region may prevent cytosolic localization. The authors should test whether the middle region (without the C-terminus) directs nuclear localization.

Extended data Fig2c. These data indicate that MCF7 cells express different isoforms of GRHL3 that show different subcellular distributions. The authors should do a Western to analyse this further. The authors should use these antibodies to test whether these isoforms are also expressed (endogeneously or after transfection) in ES cells, embryoid bodies or epidermal cells.

Fig. 2g. It is unclear what 'none' means. Are these Troma-1 negative cells? What does 'n' refer to? The authors should be more precise in what frequencies they have plotted here.

Fig. 2d-g. Apparently, deletion of any of the sequences reduces the fraction of LM cells, indicating that all domains are required for differentiation into LM. There is no correlation with the role these domains might have in controlling the subcellular distribution of the protein. In fact, adding an NES does not inflict on LM differentiation, yet removal of the CP2 DNA binding domain does. Does this mean that the DNA binding domain exerts its function outside of the nucleus?

Fig. 3g The authors claim that activation of both beta-cat and hRac1 suffices for LM cell formation. The authors need to test whether hRac1 expression alone might suffice. Quantifications of the results are required. Moreover, Rac1 is not a specific component of the PCP pathway. Nor is it clear how specific the used chemicals are in respect to PCP activation. The authors need to activate the PCP pathway by more specific means.

Fig. 3k,l The authors claim that VANGL2 does not co-localize with beta-catenin at 'considerable amounts'. The authors should show an image where cells are co-stained for VANGL2 and beta-catenin. Moreover, it is unclear whether these cells have any features of planar polarity. The authors should address this issue.

Page 10, bottom. The authors claim that pMLC is a marker for PCP activation. That is an overstatement, pMLC is often used as a proxy for mechanical forces.

Page 11 bottom The authors carefully state that the difference between Grhl3 positive and negative cells in vivo may be compatible with the difference between LP and P cells in vitro. However, differences in F-actin and pMLC might be expected between many different cell types in vivo. Additional markers need to be analysed and compared in vivo and in vitro to make a more convincing argument.

Fig. 4n,o,r,s The authors claim that Grhl3 is required for membrane localization of VANGL2 and increased levels of pMLC. How direct is this requirement? Do grhl3 mutant cells differentiate into the proper cell type?

Fig. 5d-f. The authors claim that in wt cells neither epidermal nor neural markers are expressed. However, in (d), cells identified by the arrowhead seem to express N-cadherin, which I suspect is the neural marker. The authors need to clarify this issue. I also cannot follow the authors' conclusion that in Grhl3-NLS embryos beta-catenin dependent differentiation occurred normally. What is the evidence for this?

Reference 4: There are better references than this meeting abstracts.

REF: NCOMMS-17-29972-T

“Cytoplasmic localization of GRHL3 upon epidermal differentiation triggers cell shape change for epithelial morphogenesis”

To the reviewers:

We would like to thank all three reviewers for their positive and constructive comments on the work presented in our manuscript. We have addressed most of their concerns on a point-by-point basis. However, before our detailed reply to each of the reviewers' comments, we would like to first summarize how we have addressed three major issues criticized in the previous version of our paper.

Sincerely,

Isao Matsuo, Ph.D.

Chiharu Kimura-Yoshida, Ph.D.

Major concerns

(I) *Grhl3* expression (all three reviewers)

***Grhl3* expression in LM-epidermal cells in vitro**

All three reviewers asked us to demonstrate the subcellular localization of GRHL3 in a more quantitative manner. We therefore showed the frequency of the nuclear and cytoplasmic localization of GRHL3 after 48 h (n=14) and 72 h (n=18) differentiation in culture. We have included quantitative information together with our observed confocal images in the revised version (new Fig. 3c and new Supplementary Fig. 2).

In addition, to address endogenous GRHL3 in LM-epidermal cells *in vitro*, we analyzed GRHL3 expression with two specific antibodies against mouse GRHL3 in LM-epidermal cells. Such expression studies indicated that GRHL3 localized to both the nucleus and cytoplasm of LM-epidermal cells. These new results have been included in the revised version (new Supplementary Fig. 3c–j).

As two reviewers pointed out in our initial submitted manuscript (previous Extended Data Fig. 2), domain mapping of GRHL3 fused with GFP constructs was used for subcellular localization analysis in the human breast cancer cell line, MCF7. Since the human *GRHL3* gene shows various types of transcripts (nine transcripts in total [ensembl.org/Homo_sapiens/]) while mouse *Grhl3* has a single transcript (ensembl.org/Mus_musculus/), we are not able to eliminate the possibility that mechanisms of GRHL3 localization in the human MCF7 cell line may not be identical with those in mouse ES cells and embryos. Additionally, all other experiments we

showed here were conducted in mouse epidermal cells or embryos, with the exception of domain mapping. We therefore agree with the reviewers' proposal that it is more rationale to investigate the localization of GRHL3-GFP fused protein in mouse LM-epidermal cells differentiated from ES cells. We subsequently performed and obtained almost identical results in LM-epidermal cells to those in the human MCF7 cell line (new Supplementary Fig. 3a,b). Consequently, the new results obtained using mouse LM-epidermal cells replaced all previous data obtained from the human MCF7 cell line; this included immunohistochemistry results using anti-mouse GRHL3 antibodies in the revised version (new Supplementary Fig. 3a,b).

Grhl3 expression in mouse embryos

Specific antibodies against mouse GRHL3 were newly examined using immunohistochemistry in wild-type and *Grhl3*-deficient embryos as well as a negative control (without primary antibodies; new Supplementary Fig. 5). In fact, endogenous GRHL3 expression appeared to be localized in the cytoplasm, as well as the cell membrane of surface ectoderm (SE) cells at E8.75. In addition, it also localized in the nucleus of SE cells. Moreover, protein localization of NLS-GRHL3 appeared to be increased in the nucleus of *Grhl3*^{NLS/NLS} embryos compared with that of wild-type embryos (Fig. 7h-I'). Such expression studies in mouse embryos further support our hypothesis that cytoplasmic localization of GRHL3 is crucial for neural tube closure.

(II) Interaction between *Grhl3* and planar cell polarity factors (all three reviewers)

LM-epidermal cells in vitro

Reviewer #3 pointed out that Rac1 is not a specific component of the planar cell polarity (PCP) pathway and requested us to use more specific PCP components for the induction of LM-epidermal cells. To address this, we have newly exploited a constitutively active form of mouse dishevelled associated activator of morphogenesis 1 (*Daam1*) cDNA, another crucial PCP activator (Habas et al., 2001; Nishimura et al., 2012). We subsequently transfected the constitutively active *Daam1* cDNA (*mDaam1*) together with β -catenin. Consequently, we found that LM-epidermal cells were induced by *mDaam1* together with β -catenin while neither *mDaam1* nor β -catenin alone induced LM-epidermal cells (new Fig. 4a,h,i). This finding indicates that the induction of LM-epidermal cells involves both *mDaam1* and β -catenin.

Reviewer #1 was also concerned that LM-epidermal cells were really induced via the PCP pathway because LM-epidermal cells were still formed to some extent (about 70%) in the presence of chemical inhibitors. However, we think that even if we had used *DN-Grhl3*, this would have been unable to inhibit the formation of *Grhl3*-dependent LM-epidermal cells to 0%; 30% of embryoid bodies (EBs) continued to form LM-epidermal cells (new Fig. 1g). Thus, since we always found all EBs formed LM-epidermal cells by *Grhl3* cDNA expression, we think that it is very important to emerge EBs having no LM-epidermal cell.

In addition, we newly tested another chemical inhibitor of Rac1, NSC23766, and

found that induction of LM-epidermal cells was repressed to less than 30% (new Fig. 4g). These additional results are now described in text and figure panels together with statistical data, which were requested by all reviewers, and have been corrected appropriately (new Fig. 4g).

Finally, we newly examined expression of SCRIB, another PCP molecule in the LM-epidermal cells and found that SCRIB products were exclusively induced in the cytoplasm of *Grhl3*-dependent LM-epidermal cells (new Fig.5c).

These above new findings altogether further strengthen our hypothesis that LM-epidermal cells are induced through at least PCP-related molecules. However, as reviewer #3 pointed out, given that planar cell polarity-specific features are still unclear in LM-epidermal cells and that it is very difficult to exclude the possibility that LM-epidermal cells may be induced except via the PCP pathway, we would like to soften our statement regarding the induction of LM-epidermal cells via the PCP pathway throughout the manuscript and change this to “non-canonical Wnt pathway or PCP-related molecules” in the revised text in place of “the PCP pathway” in the original text.

Mouse embryos

Since reviewers asked us to analyze additional markers specific to the PCP pathway, we have examined embryos with three additional markers, CELSR1, RHOA, SCRIB, via specific antibodies. We have newly shown in this revised version that these proteins can be found in the cell membrane and cytoplasm of wild-type embryos (new Fig. 6g–j) and were absent in *Grhl3*-deficient embryos (new Fig. 6g’–j’). Moreover, expressions of VANGL2, CELSR1 and RHOA were apparently reduced in *Grhl3*^{NLS/NLS} embryos (new Fig. 8g–k,h–l). Therefore, cytoplasmic localization of GRHL3 is essential for the correct expression of these PCP related–molecules in SE cells during neural tube closure. These additional data are consistent with our genetic data that *NLS-Grhl3* interacts with *Vangl2*, but not *β-catenin*, for mouse neural tube closure (Fig. 8o).

In addition, as reviewer #1 pointed out, *Rhogef19* mRNA has been shown to be one of the direct transcriptional targets of *Grhl3* (Caddy et al., 2010). So, we newly examined whether NLS-GRHL3 could affect *Rhogef19* mRNA expression at the transcriptional level. In fact, *Rhogef19* mRNA was lost in *Grhl3*-deficient embryos (new Fig. 8p,q,s,t). Moreover, we newly found that *Rhogef19* mRNA could still be found in *Grhl3*^{NLS/NLS} embryos (new Fig. 8r,u). These new findings further support the notion that nuclear-localized GRHL3 causes defects of neural tube closure not via transcriptional *Rhogef19* expression.

The above additional data together further strengthen our hypothesis that nucleus-localized GRHL3 is not sufficient for epithelial morphogenesis such as neural tube closure and wound repair that are mediated by molecules involving the PCP pathway.

With respect to the molecular mechanisms of GRHL3 involved in the control of expression of PCP-related molecules, we are only now starting to identify proteins that interact with GRHL3 and analyzing their functions. Identification and characterization of these related proteins will clarify how GRHL3 regulates PCP proteins in future. Since the novelty of our work relates to revealing how cytoplasmic-localized, but not nuclear-localized type of GRHL3

is crucial for cell shape change processes during neural tube closure *in vivo*, we would like to kindly ask the editors and reviewers to consider that clarification of the precise molecular mechanisms of GRHL3 controlling the expression of PCP-related molecules be regarded as beyond the scope of this manuscript.

(III) *In vivo* counterpart of LM-epidermal cells (Reviewers #1 & #3)

Since we did not undertake *Grhl3* misexpression experiments in mouse embryos in our previous manuscript, we have newly carried out two kinds of misexpression experiments by means of the generation of transgenic mice and local electroporation: the early ubiquitous expression of *Grhl3* in fertilized eggs under the control of the chicken β -actin promoter (CAG-promoter) and the local expression of *Grhl3* in SE at E8.5 via *ex utero* electroporation. These two series of experiments clearly indicated that *Grhl3* overexpression up-regulated F-actin expression and appeared to induce multinucleation (new Fig. 2). Moreover, PCP-related molecules were also up-regulated in mouse embryos by *Grhl3* cDNA (new Fig. 6n-o’).

With respect to the polyploidy-like phenotype induced by *Grhl3*, it is well known that the epidermis of mammalian skin becomes polyploidy during homeostasis (Zanet et al., 2010; Orr-Weaver, 2015, Gandarillas & Freije, 2014). In addition, wound-induced polyploidization of the epidermis has also only recently been discovered and analyzed in the mouse (Losick et al., 2016); epidermal cells after injury are multinucleated and become enlarged in size in order for epithelial structure to recover quickly. As *Grhl3* is also crucial for skin development as well as wound repair processes (Caddy et al., 2010), it may be rational that *Grhl3* contribute to multinucleation and the enlargement of cell size during normal development as well as wound repair. However, we would like that any detailed study of the connection between polyploidization and *Grhl3* not be considered the main focus of the current study and but rather be thought of as an interesting subject for further investigation.

These several lines of evidence, together with our new misexpression findings *in vivo*, clearly suggest that the LM-epidermal cells observed to be induced by *Grhl3* suggest these are not *in vitro* artifacts specific to ES differentiated epidermal cells but rather similar phenotypes that are observed in mouse embryos *in vivo*.

Reviewers' comments:**Reviewer #1**

The manuscript by Yoshida et.al proposes a new mechanism by which the nuclear transcription factor Grhl3 transitions from the nucleus to the cytoplasm during embryogenesis triggering epithelial morphogenesis. The paper is potentially significant because it proposes a non-transcriptional mechanism by which Grhl3 is required for planer cell polarity during embryogenesis. Grhl3 has been previously linked to planar cell polarity but through transcriptional mechanisms. Therefore, this study proposes a new, radically different mechanism. The authors present experiments with mice (Grhl3^{ns1/ns1}) where they knocked in an extra nuclear localization signal into the Grhl3 transcript and showed that a phenotypic abnormality (defect in neural tube closure) that is similar, however milder, than what was observed in Grhl3 deleted mice Grhl3^{-/-}. Presumably, Grhl3 does not localize to the cytoplasm in this mouse. This supports the proposed model where cytoplasmic translocation of Grhl3 during embryogenesis is required.

I) The weakness of the paper is that most of the figures deal with an in vitro model where overexpression of Grhl3 leads to the formation of big, multinucleated cells, apparently in response to cytoplasmic localized Grhl3. The problem is that these cells seem highly artificial and the authors never show such cells in vivo. This is an interesting finding but because of it is an in vitro cell culture phenomenon, it is hardly of much general interest. The mouse experiments are stronger but they also have some weakness as discussed below.

Response: Please read the response to major concern (III) above.

II) Another issue with this paper is the writing. It is difficult to understand how the authors use the term morphogenesis. Conventionally, this term is used to describe the formation of an organism, an organ or a part of an organ. As such morphogenesis usually depends on multiple cellular features such as proliferation, migration, differentiation, etc. Therefore, it is strange to state “triggers epithelial morphogenesis from differentiation”. It appears that the authors use the term to describe cell shape changes, which is not traditional. Also, since Grhl3 has a role in the terminal differentiation of the interfollicular epidermis during embryogenesis, the authors might make it more clear upfront (at least in the abstract) that when they are talking about epidermal differentiation, they are talking about the early commitment to the surface epidermis.

Response: Following the reviewer’s suggestion, we have corrected the meaning of the term “morphogenesis”, which includes proliferation, migration, differentiation, in addition to cell shape change. Accordingly, the title of this manuscript is amended as follows “Cytoplasmic localization of GRHL3 upon epidermal differentiation triggers cell shape change for epithelial morphogenesis”. In addition, we have changed the use of the word “morphogenesis” throughout the manuscript to more specifically mean “cell shape change”.

Other comments:

1. In Figure 1, the authors noticed the large mature epidermal cells induced by Grhl3 cytoplasmic translocation which don't seem to have biological relevance in live animals during development. The authors should apply experimental controls to eliminate artifacts that are induced by in-vitro transfection (over-expression of Grhl3). In addition, overexpressing Grhl2, doesn't mean is endogenously expressed in EB and p-epidermal cells (as stated by the authors in the text).

Response: Please read the response to major concern (III) above as our response to the first part of the above comment.

Regarding the *Grhl2* misexpression experiment, we agree with the reviewer’s comment that “*Grhl2* is endogenously expressed in embryoid bodies (EBs) as well as the periphery or outside of EBs and overexpression of *Grhl2* might not be meaningful”. So, we transfected the cDNA of *Grhl1*, which is not expressed endogenously like *Grhl3* in EBs as well as neurula embryos, and found that *Grhl1* cDNA did not induce LM-epidermal cells efficiently

(new Fig. 1d,g). In addition, a dominant negative form of the *Grhl3* construct (*DN-Grhl3*) was transfected together with *Grhl3* cDNA. Consequently, *DN-Grhl3* prevented the induction of LM-epidermal cells induced by *Grhl3* cDNA (new Fig. 1f,g). These two types of misexpression studies clearly indicate that the *Grhl3* cDNA specifically induced LM-epidermal cells.

2. In Figure 2, the authors should include experimental statistics (number of biological replicates showing halotag-grhl3 cytoplasmic translocation, number of biological replicates showing reduction in LM epidermal size in CAG-NLS-Grhl3 transfected cells).

Response: Please read the response to major concern (I) above. In accordance with the reviewer's suggestion, we counted the number of LM-epidermal cells showing nuclear or cytoplasmic localization of GRHL3 and described the frequency in the revised version (new Fig. 3c and Supplementary Fig. 2). In addition, biological replicates of numbers of LM-epidermal cells by *CAG-NLS-Grhl3* cDNA were indicated in the previous Fig. 2j (now Fig. 3i). Since the number of LM-epidermal cells by *CAG-NLS-Grhl3* was clearly reduced, as can be observed in the new Fig. 3i, the size of LM-epidermal cells was not analyzed in previous and current versions of the manuscript.

3. In Figure 3, the authors claim that activation of PCP is necessary for LM-epidermal cells formation and inhibiting PCP blocks Grhl3-dependent LM epidermal cells formation. The authors don't show clear PCP activation exclusively in LM-epidermal cells compared to other cells in the EB (expression of PCP markers). When PCP was inhibited (CAG-Dsh-DEP), LM epidermal cells were still formed (~ 70% of total EBs), this data dispute the authors claim. In addition, expression of Grhl3 and Vangl2 in LM epidermal cells cytoplasm is not enough data to conclude that grhl3 contributes to PCP via vangl2.

Response: Please read the response to major concern (II) above. In addition, our intricate figure panels may have misled the reviewer (previous Fig. 3). So, in order to avoid confusion, we have corrected all these panels in the revised version (new Fig. 4).

4. In Figure 4, the authors claim that Grhl3 deletion led to abolished Vangl2 protein expression, however, the provided IF images show normal staining of Vangl2 in Grhl3-/- embryos. Also, the authors need to be consistent in IF images magnifications (Figure 4 a-e).

Response: We had omitted labeling the genotype of “p and q” panels in the previous Fig. 4. Consequently, the reviewer mentioning that “the provided IF images show the normal staining of VANGL2 in *Grhl3*^{-/-} embryos” which may be due to the reviewer misconstruing the VANGL2 expression of the *Grhl3*^{cre/+} heterozygous embryo (previous Fig. 4r,s; now Fig. 6e,f). However, no VANGL2 expression was observed in *Grhl3*^{cre/cre} embryos (previously Fig. 4r,s and now Fig. 6g,g'). To avoid such a confusion in readers, we have correctly labeled genotypes in the revised version (now Fig. 6g,g'). Also in accordance with the reviewer's suggestion, we have modified the magnifications of images (new Fig. 6).

It is surprising that staining with Grhl3 antibody show no fluorescence background in Grhl3-/- embryos. The authors must include negative control (no antibody) staining WT embryos to validate Grhl3 antibody staining.

Response: Please read the response to major concern (I) above. Our additional new data has validated several antibodies against mouse GRHL3 used in this manuscript (new Supplementary Fig. 5).

5. In Figure 5, the authors have generated a mouse model where *Grhl3* expression is restricted only to the nucleus. The authors have to show that *Grhl3* is normally expressed in these embryos and that *Grhl3* is restricted to the nucleus by staining for *Grhl3* in *Grhl3^{NSL/NSL}*, *Grhl3*^{-/-} and wild type embryos where *Grhl3* is expected to be cytoplasmic like claimed.

Response: Please read the response to major concern (I) above.

To address the reviewer's point, we have newly shown GRHL3 expression in wild-type and *Grhl3*-deficient embryos with immunohistochemistry using three different polyclonal antibodies in the revised version of the manuscript (Supplementary Fig. 5a–c'). GRHL3 expression appeared to be observed in the nucleus and cytoplasm, including the cell membrane of surface ectoderm cells. The antibody (aa49–134) recognized GRHL3 in the cytoplasm and partly in the nucleus (Supplementary Fig. 5a,a'). The other two antibodies (aa195–211, aa478–493) recognized GRHL3 mainly in the cytoplasm and cell membrane on the lateral side of surface ectoderm, and partly in the nucleus in cells of neural folds, the most medial side (Supplementary Fig. 5b–c', new Fig.10). In addition, immunohistochemical analysis with the antibody (aa478–493) indicated that GRHL3 localization in *Grhl3^{NSL/NSL}* embryos apparently shifted from the cytoplasm to the nucleus (Fig. 7h–i'). These findings together explicitly demonstrate that GRHL3 is able to localize in the cytoplasm and cell membrane in addition to the nucleus.

*The authors have to elucidate that the phenotype observed in (*Grhl3^{NSL/NSL}*) is not due to alteration in the ability of *Grhl3* to transcriptionally regulate genes involved in neural tube closure and epidermal differentiation.*

Response: Please also read the response to major concerns (II) above.

We have newly analyzed *Rhogef19* mRNA expression using *in situ* hybridization: *Rhogef19* mRNA is a direct transcriptional target of GRHL3 and is down-regulated in *Grhl3*-deficient embryos as previously demonstrated (Fig. 8p,q,s,t; Caddy et al., 2010; Darido and Jane, 2010). In contrast, *Rhogef19* mRNA expression appeared to not be reduced in *Grhl3^{NSL/NSL}* embryos (Fig. 8r,u). Our new findings clearly support the notion that the transcriptional activities of GRHL3 fused with NLS in *Grhl3^{NSL/NSL}* embryos are not defective.

*The authors claim that *Grhl3* is necessary for correct membrane localization of VANGL2 at the protein levels based on IF staining in vivo, did the authors observe similar phenotype in vitro if *Grhl3* is mutated?*

Response: In *in vitro* embryoid bodies where *Grhl3* was misexpressed, VANGL2 expression was only found in GRHL3-positive LM-epidermal cells but not in other non-LM-epidermal cells (previously Fig. 3k; now Fig. 5a,b; please see attached data 1 & data 3 below). These findings strongly suggest that GRHL3 expression is essential for VANGL2 expression in LM-epidermal cells *in vitro*. Thus, since mutations of *Grhl3* fail to form LM-epidermal cells appropriately, it is technically impossible to analyze VANGL2 expression in epidermal cells *in vitro*.

Attached data1 VANGL2 expression was only detectable in LM-epidermal cells. An entire single embryoid body is shown after transfection with *Grhl3* cDNA and HaloTag-VANGL2, and immunohistochemical staining for TROMA-1 and HaloTag antibodies, and DAPI (Left). A single image of HaloTag-VANGL2 expression is shown (right).

*In addition, how did the authors determine that wound healing was delayed in *Grhl3^{NSL/NSL}* embryos? The author should state the parameters of wound healing measurements and include a figure with quantifications and biological replicates.*

Response: We showed that injured epidermal structures were not repaired in *Grhl3^{NSL/NSL}* embryos after 24 h-culture *in vitro*; however, “delayed” was incorrectly used in the original text. So, we have amended the word “delayed” in the revised version to “*Grhl3^{NSL/NSL}* embryos did not show repair of the hind-limb amputation unlike wild-type embryos”. To describe the state of wound repair, we took scanning electron micrographs and now present new data together with biological replicates in the revised version of the manuscript (new Supplementary Fig. 7).

*6. In Figure 6 the authors claim that cytoplasmic localization of *Grhl3* is required for PCP and mechanical strength, the authors should include *Grhl3-NSL* embryos in the experiment to support the proposed hypothesis.*

Response: We have newly calculated the mechanical strength in *Grhl3^{NSL/NSL}* embryos with an atomic force microscope (AFM). These data have been included in the revised version of the manuscript (page 18, line 371). In addition, we have newly analyzed the expression of the PCP-related molecules, CELSR1 and RHOA, in *Grhl3^{NSL/NSL}* embryos and found both of the two markers were reduced (Fig. 8i–l). These additional results further strengthen our hypothesis on the crucial roles of cytoplasmic-localized GRHL3.

In addition, the technique using whole embryo to measure the mechanical strength doesn't exclude the involvement of other non-epithelial cells in the data collection. The authors must use appropriate controls for the experiment.

Response: This reviewer casts doubt on whether the mechanical strength measured by an AFM correctly corresponds to that of surface ectoderm. Generally, if the depth of indentation of the pushed AFM cantilever was greater than 10% of the surface ectoderm cell height, inner tissues such as mesenchymal tissues affect the value of the mechanical strength measured. However, in this study, indentations of the AFM cantilever were from 21 nm to 578 nm in depth (wild type,

135 nm; *Grhl3*^{cre/cre}, 323 nm; *Grhl3*^{NLS/NLS} 38.7 nm on average) while the height of surface ectoderm cells was about 20 μm , (from 15.6 to 24.8 μm). Given that the depth of indentation by the cantilever was less than 10% of the height of surface ectoderm cells, the mechanical strength measured in this series of experiments is evaluated according to the physical properties of mainly surface ectoderm cells and are unlikely to be affected by other non-epithelial cells such as inner mesenchymal tissues.

7. Lastly, Grhl3 is known to regulate PCP on the transcriptional level by directly binding to RhoGEF19 promoting actin polymerization, the authors need to show that this is not the case here and that Grhl3NSL/NSL doesn't alter the expression of RhoGEF19 or polymerization of actin.

Response: Please read the above response to major concern (II) and the specific comments of this reviewer's question "5" above.

Reviewer #2

In this manuscript Kimura-Yoshida et al explore the role of GRHL3 in epidermal specification. The authors find that in addition to its previously reported role in b-catenin mediated specification of epidermal fate, GRHL3 is also important for epidermal maturation via modulation of PCP signalling. Most notably the authors show in vivo that the consequence of these interactions is a change in the biomechanical properties of the epithelia.

This is a very interesting manuscript that brings new light into how the epidermis differentiates. The manuscript would however greatly benefit from better quantification of a number of assays and a clearer description of some of the data. Some of these issues are detailed below:

1) The introduction lacks a lot of background detail regarding epidermal cells specification and differentiation. Also, some background on what is different between LM- and P-epidermal cells would be good.

Response: In accordance with the reviewer's suggestion, we have added a paragraph introducing the concept of epidermal specification during skin development in the revised version of the manuscript (pages 3-4, lines 55-71).

In addition, our definition of P-epidermal cells is that they form in the surrounding or periphery of the embryoid body colony without any additional factors (Fig. 1a,b), while LM-epidermal cells form on or within the central embryoid body colony in response to *Grhl3* or other factors. Thus, these two cell types are judged according to the location of formation and cell size. Since P-epidermal cells are induced without any additional factors in our *in vitro* culture conditions and the formation of these cells is not the main subject of this study, we have removed all statements regarding the formation of P-epidermal cells from the text of the revised version of the manuscript.

2) Figure 1f – The authors argue that a dominant-negative form of Grhl3 cDNA (DN-Grhl3), reduces the number of LM-epidermal cells (Fig. 1e,f), but in both control transfected and DN-Grhl3 transfected EBs this number appears to be close to 0, therefore it is not possible to conclude that the DN inhibits LM-epidermal differentiation.

Response: We agree with the reviewer's point. To address this, we newly transfected *DN-Grhl3* together with *Grhl3* into ES cells. Consequently, *DN-Grhl3* reduced the number of LM-epidermal cells induced by *Grhl3* to less than 30% (new Fig. 1f,g). This new finding has been included in the revised version of the manuscript.

3) Figure 2a-b- The authors need to provide quantification of what proportion of cells have nuclear and cytoplasmic GRHL3 localization at 48 and 72 hours post-transfection.

Response: Please read the above response to major concern (I) above. In line with this suggestion, we counted the number of LM-epidermal cells showing nuclear or cytoplasmic localization of GRHL3 and described the frequency in the revised manuscript (new Fig. 3c and Supplementary Fig. 2).

4) The labelling of some of the figures is not clear, for example in Figure 3a-h, it takes a while to work out which treatment the cells received

Response: In response to this comment, we modified the panels in question in the revised version of the manuscript (now Fig. 4). We think these corrections will avoid confusing readers.

5) Extended Figure 3b. The authors need to quantify the ability of beta-catenin to induce LM-epidermal cells.

Response: In response to the reviewer's suggestion, we have included data regarding the induction of LM-epidermal cells by β -catenin alone in the revised manuscript (new Fig. 4a; 0%, n=31).

6) Figure 3 and page 9 of the text appear to contradict each other. The text suggests that PCP activity is modulated independently of Grhl3, but the labelling of the figure suggests otherwise. The authors need to clarify with proper quantification which are the effects that manipulating PCP alone has on LM-epidermal cell differentiation and which effects does manipulating PCP and over-expressing Grhl3 have on this differentiation. Similarly the effects that b-catenin and PCP manipulation alone and together need to be properly quantified.

Response: Please read the response to major concern (II) above. Since the text and data panels relating to PCP modulation appear to be confusing, we have completely re-written the corresponding text together with figure panels in the revised version of the manuscript (new Fig. 4 and its legends).

7) Figure 3a-b. The authors need to quantify the effects of PCP modulation alone on LM-induction.

Response: In response to this suggestion, we have included quantitative data on the induction of LM-epidermal cells in the revised manuscript (Fig. 4 and its legends).

8) Figure 3k-m. The authors need to provide quantification for the different co-localization experiments.

Response: In line with the reviewer's suggestion, we have shown the number of colonies showing mKG co-localization in the revised manuscript (n=17, new Supplementary Fig. 4d). Additionally, we have added panels on confocal signal intensities showing VANGL2 and GRHL3 distributions to the revised manuscript (new Fig. 5b). These additional data support our statement that some GRHL3 and VANGL2 appears to co-localize in LM-epidermal cells. Although we were unable to analyze VANGL2 and GRHL3 localization simultaneously with antibodies in mouse embryos since both antibodies were generated as rabbit polyclonals, both GRHL3 and VANGL2 appear to localize in the cell membrane of the surface ectoderm of wild-type mouse embryos (Fig. 6f,g, Supplementary Fig. 5b,c). These findings together suggest GRHL3 and VANGL2 are present in close proximity in LM-epidermal cells and the embryonic surface ectoderm. However, we think that GRHL3 is unlikely to bind VANGL2 directly because yeast two-hybrid experiments with GRHL3 and VANGL2 were negative (our unpublished observations).

To further clarify molecular mechanisms of GRHL3 controlling expression of PCP-related molecules, we are currently identifying proteins that interact with GRHL3 and have commenced analyzing their function. Thus, identification of these related proteins will clarify how GRHL3 regulates the VANGL2 protein in future. Since the primary novel aspect of our work is describing how cytoplasmic-localized but not nuclear-localized GRHL3 is crucial for cell shape change processes during neural tube closure *in vivo*, we would like to kindly ask the editor and reviewers to consider that clarification of the precise molecular mechanisms of GRHL3 controlling PCP-related molecules is beyond the scope of this manuscript.

Reviewer #3

The manuscript by Kimura-Yoshida addresses the role of Grhl3 in epithelial morphogenesis by combining in vitro and in vivo approaches. In vitro, the authors show that forced Grhl3 expression can induce large epidermal cells that are enriched in actomyosin. Induction of these large epidermal cells appears to require cytoplasmic Grhl3 and can be blocked by a dominant-negative version of Dsh. In vivo, a mutant form of Grhl3 that is fused to NLS interacts with Vangl in double heterozygous animals, resulting in spina bifida in about a third of the embryos. Double heterozygous of Grhl3-NLS and beta-catenin did not show this phenotype. Finally, the authors use micropipette aspiration assays and AFM to measure and compare mechanical properties in Grhl3 expressing and non-expressing cells in vivo and in vitro. The authors conclude that the cytoplasmic pool of Grhl3 activates the PCP pathway to allow epidermal cells to acquire mechanical properties important for morphogenesis.

The manuscript addresses an interesting and important question with regard to the relationship between cell differentiation and morphogenesis. The strength of the manuscript is the use of different mouse mutants to test for genetic interactions. However, as outlined in detail below, a number of claims by the authors are not well supported. The authors central claim that cytoplasmic Grhl3 controls morphogenesis, for example, and the link between Grhl3 and PCP are weak. Moreover, there is little mechanistic understanding of how cytoplasmic Grhl3 could control morphogenesis. The conceptual advance of this work is therefore somewhat limited.

Specific comments

Fig. 1b-e.

Control and experiment are difficult to compare. The authors should show images at the same magnification and stained for the same markers.

Response: With regard to the reviewer's suggestion, we have corrected the panels of the previous Fig. 1b–e in the revised manuscript (now Fig. 1b–f).

It is unclear how the authors distinguish between P-epi and LM-epi. Simply by size?

Response: Our definition of P-epidermal cells is that they form in the periphery or outside of the embryoid body colony without any additional factors (Fig. 1a,b), while LM-epidermal cells form on or within the central embryoid body colony in response to *Grhl3* or other factors. Thus, we have judged these on the basis of the location of formation and cell size as well as multinucleation. In addition, since P-epidermal cells are induced without any additional factors under our *in vitro* culture conditions and the formation of P-epidermal cells is not the subject of this study, we have removed all statements regarding P-epidermal cells from the text in the revised manuscript.

What is the transfection efficiency? Do all cells express Grhl3 or only those cells that differentiate into LM cells? The authors should include a marker for transfected cells.

Response: The transfection efficiency appears to be very high as shown below (attached data 2). However, we cultured transfected ES cells with neomycin selection for 48 h and made embryoid bodies. When we analyzed GRHL3 expression in the cultured embryoid bodies, most GRHL3-positive cells became LM-epidermal cells as we have shown below (attached data 3). Therefore, it is highly likely that most GRHL3-positive cells differentiate into LM-epidermal cells.

Attached data2 Transfection efficiency after lipofection of the CAG-EGFP plasmid into embryonic stem (ES) cells. Bright-field observation of ES cells after 48 h Transfection (left). GFP fluorescence images (right). Most ES cells expressed GFP.

Attached data 3 Most GRHL-positive cells appeared to differentiate into LM-epidermal cells. The entire single embryoid body is shown after transfection of *Grhl3* cDNA. Immunohistochemical staining using antibodies against GRHL3 and TROMA-1, and DAPI.

What is the in vivo counterpart of LM-cells?

Response: Please read our response to the major concern (III) above.

Fig. 1f.

I am puzzled that Grhl3 induces 100% LM cells. In Fig1c, it appears that only a small fraction of EB cells differentiated into LM cells. The authors need to clarify.

Response: 100% LM-epidermal cells means that all embryoid bodies we observed had at least more than one LM-epidermal cell. For example, if we analyze 18 embryoid bodies, 100% indicates that all 18 embryoid bodies have at least more than one LM-epidermal cell. A value of 50% indicates nine embryoid bodies among a total of 18 embryoid bodies have more than one LM-epidermal cell and that the remaining nine embryoid bodies do not have any LM-epidermal cells.

Fig. 1e,f.

The authors claim that DN-Grhl3 reduces the number of P-epidermal cells and refer to Fig. 1f. However, Fig1f details the number of LM, but not P epidermal cells. The authors should show a separate quantification of P epidermal cells.

Response: As we have mentioned in the above comments regarding Fig. 1b–e, we have removed all statements regarding P-epidermal cells from the text in the revised manuscript.

Fig. 1b-b", d. The authors claim that GRHL2 is endogeneously expressed in EBs as well as P-epidermal cells and refer to Fig. 1b-b",d. However, these Figs do not show GRHL2 staining. The authors' conclusion at the end of this paragraph that both Grhl2 and Grhl3 are able to contribute to epidermal contribution are not supported by data.

Response: *Grhl2* is endogenously expressed in embryoid bodies but we have now removed the *Grhl2* cDNA misexpression data from the manuscript. Instead, we conducted a new misexpression experiment using *Grhl1* cDNA to identify the specific role of *Grhl3* in the revised manuscript (new Fig. 1d–g).

Page 6. *"Since the transcriptional activity of CP2 transcription factors..." How is this related to Grhl3?*

Response: We have amended the sentence to make clear that *Grhl3* belongs to the CP2 transcription factor family in the revised version of the manuscript.

Fig. 2/ Page 6

The authors should stress that GRHL3 expression is analysed in GRHL3 transfected cells.

Response: In line with this comment, we have added new data regarding GRHL3 expression in *Grhl3* transfected cells in the revised manuscript (Supplementary Fig. 2,3).

Page 7, top.

The authors should mention that the analysis of the localization of GRHL-3 subdomains was done in MCF7 cells, a breast cancer cell line. Could the authors transfect these GFP-tagged subdomains into ES cells and then analyse subcellular localization in epidermal cells?

Response: Please read the response above to major concern (I). Yes, we analyzed subcellular localization in LM-epidermal cells in the revised manuscript (new Supplementary Fig. 3b).

Extended Fig. 2.

The authors claim that GRHL3 localized to the cytoplasm through N and C termini, and to the nucleus through the middle region. However, this Fig. shows that the C-terminus directs both cytoplasmic and nuclear localization. The middle region may prevent cytosolic localization. The authors should test whether the middle region (without the C-terminus) directs nuclear localization.

Response: Please read our response to major concern (I) above.

Following the reviewer's suggestion, we generated a new construct carrying solely the CP2 domain of *Grhl3* and analyzed protein localization in LM-epidermal cells (new Supplementary Fig. 3b). Consequently, we found that CP2 domains appear to direct localization to both the cytoplasm and nucleus.

Extended data Fig2c.

These data indicate that MCF7 cells express different isoforms of GRHL3 that show different subcellular distributions. The authors should do a Western to analyse this further. The authors should use these antibodies to test whether these isoforms are also expressed (endogenously or after transfection) in ES cells, embryoid bodies or epidermal cells.

Response: Please read the response to major concern (I) above.

Fig. 2g. *It is unclear what "none" means. Are these Troma-1 negative cells? What does "n" refer to? The authors should be more precise in what frequencies they have plotted here.*

Response: The word "none" indicates no TROMA-1-positive cells were observed in embryoid bodies. In order to avoid confusion, we describe typical examples of the criteria for epidermal cells in the revised paper (Supplementary Fig. 1h–k). The "n" indicates the number of embryoid bodies we analyzed. Thus, in the case of the furthest left side bar of the previous Fig. 2g (now Fig. 3f), which corresponded to *Grhl3* transfection, we observed that all 21 embryoid bodies had more than one LM-epidermal cell in a total of 21 embryoid bodies we analyzed; this corresponds to a value of 100% (one typical LM-epidermal image is also shown in Supplementary Fig. 1h showing one embryoid body colony).

Fig. 2d-g.

Apparently, deletion of any of the sequences reduces the fraction of LM cells, indicating that all domains are required for differentiation into LM. There is no correlation with the role these domains might have in controlling the subcellular distribution of the protein. In fact, adding an NES does not inflict on LM differentiation, yet removal of the CP2 DNA binding domain does. Does this mean that the DNA binding domain exerts its function outside of the nucleus?

Response: We have carefully re-analyzed NES-fused GRHL3 localization in LM-epidermal cells and found that this fusion protein still localized to the nucleus, but to a lesser extent (please see the below attached data 4). Therefore, the residual nuclear localization of GRHL3 may contribute to epidermal differentiation.

In addition, deletion of the CP2 DNA-binding domain indicates the highest frequency of the emergence of the TROMA-1 negative "none" phenotype. Thus, the CP2 domain appears to be highly crucial for epidermal differentiation, which includes all types of TROMA-1-positive epidermal cells.

Attached data 4 NES-fused GRHL3 expression is still found in the nucleus, but to a lesser extent.

Fig. 3g *The authors claim that activation of both beta-cat and hRac1 suffices for LM cell formation. The authors need to test whether hRac1 expression alone might suffice. Quantifications of the results are required.*

Response: We have added a panel showing *hRac1* cDNA alone in the revised version since LM-epidermal induction was 0% (n=7; Fig. 4j).

Moreover, Rac1 is not a specific component of the PCP pathway. Nor is it clear how specific the used chemicals are in respect to PCP activation. The authors need to activate the PCP pathway by more specific means.

Response: Please read the response to major concern (II) above.

Fig. 3k,l *The authors claim that VANGL2 does not co-localize with beta-catenin at 'considerable amounts'. The authors should show an image where cells are co-stained for VANGL2 and beta-catenin.*

Response: We apologise for this oversight. We have intended and corrected this to “**GRHL3** does not co-localize with β -catenin in considerable amounts” in the revised paper.

Moreover, it is unclear whether these cells have any features of planar polarity. The authors should address this issue.

Response: Please read our response to major concern (II) above.

Page 10, bottom

The authors claim that pMLC is a marker for PCP activation. That is an overstatement, pMLC is often used as a proxy for mechanical forces.

Response: Please read the response to major concern (II) above. Addressing the reviewer's concern, we have corrected our statement regarding pMLC in the text of the revised manuscript.

Page 11 bottom

The authors carefully state that the difference between Grhl3 positive and negative cells in vivo may be compatible with the difference between LP and P cells in vitro. However, differences in F-actin and pMLC might be expected between many different cell types in vivo. Additional markers need to be analysed and compared in vivo and in vitro to make a more convincing argument.

Response: Please read the response to major concern (II) and (III) above.

Fig. 4n,o,r,s

The authors claim that Grhl3 is required for membrane localization of VANGL2 and increased levels of pMLC. How direct is this requirement? Do grhl3 mutant cells differentiate into the proper cell type?

Response: With respect to the precise molecular mechanisms of VANGL2 localization by *Grhl3*, please read the response to major concern (II) above.

In our previous work (Kimura-Yoshida et al., 2015), we showed that surface ectoderm cells of *Grhl3*-deficient embryos failed to differentiate into epidermal cells correctly but rather differentiated into neural ectoderm (Fig. 6D–E' of Kimura-Yoshida et al., 2015). Additionally, given that lateral portions of the surface ectoderm express *Grhl2* during neural tube closure, with the exception of the neural plate border (please see our previous paper; Fig. 5

of Kimura-Yoshida et al., 2015), the roles of *Grhl3* in epidermal differentiation, i.e. transcriptional function, may be complemented by redundant *Grhl2* expression (please see our previous paper; Fig. S7 of Kimura-Yoshida et al., 2015).

Fig. 5d-f. *The authors claim that in wt cells neither epidermal nor neural markers are expressed. However, in (d), cells identified by the arrowhead seem to express N-cadherin, which I suspect is the neural marker. The authors need to clarify this issue. I also cannot follow the authors' conclusion that in Grhl3-NLS embryos beta-catenin dependent differentiation occurred normally. What is the evidence for this?*

Response: In the previous Fig. 5d (now Fig. 8d), neural plate border cells that were located between the surface and neural ectoderms expressed neither Troma-1 nor N-cadherin as previously shown (please see our previous paper; Fig. 1 of Kimura-Yoshida et al., 2015). Given that Fig. 8d–f are reconstructed 3D images, nuclei stained with DAPI (colored in green) were always invisible from the outside when cells expressed N-cadherin (colored in dark blue) at the cell membrane (now Fig. 8e), while nuclei stained with DAPI were visible from the outside of embryos when cells did not express N-cadherin at the cell membrane (now Fig. 8d [wild type], 8f [*Grhl3^{NLS/NLS}*]). In addition, we have newly analyzed *Rhogef19* mRNA expression with *in situ* hybridization: *Rhogef19* expression appeared to be unchanged in *Grhl3^{NLS/NLS}* embryos (Fig. 8r,u). This new finding clearly supports our hypothesis that transcriptional activities of GRHL3 fused with NLS in *Grhl3^{NLS/NLS}* embryos can work well.

In addition, we have previously shown that such epidermal differentiation processes from uncommitted ectodermal progenitors between neural and surface ectoderms is β -catenin–dependent by means of conditional β -catenin knock-out experiments (please see our previous paper; Fig. 8 of Kimura-Yoshida et al., 2015).

Please also read the response to major concern (II) above.

Reference 4: *There are better references than this meeting abstracts.*

Response: We agree and apologise for this oversight. We have corrected the reference in the revised paper (new references #4-7 , page 28).

[References]

Caddy J, Wilanowski T, Darido C, Dworkin S, Ting SB, Zhao Q, Rank G, Auden A, Srivastava S, Papenfuss TA, Murdoch JN, Humbert PO, Parekh V, Boulos N, Weber T, Zuo J, Cunningham JM, Jane SM. Epidermal wound repair is regulated by the planar cell polarity signaling pathway. *Dev Cell*. 2010 Jul 20;19(1):138-47.

Darido C, Jane SM. Grhl3 and GEF19 in the front rho. *Small GTPases*. 2010 Sep;1(2):104-107.
Gandarillas A, Freije A. Cycling up the epidermis: reconciling 100 years of debate. *Exp Dermatol*. 2014 Feb;23(2):87-91.

Habas R, Kato Y, He X. Wnt/Frizzled activation of Rho regulates vertebrate gastrulation and requires a novel Formin homology protein Daam1. *Cell*. 2001 Dec 28;107(7):843-54.

Kimura-Yoshida C, Mochida K, Ellwanger K, Niehrs C, Matsuo I. Fate Specification of Neural Plate Border by Canonical Wnt Signaling and Grhl3 is Crucial for Neural Tube Closure. *EBioMedicine*. 2015 Apr 18;2(6):513-27.

Losick VP, Jun AS, Spradling AC. Wound-Induced Polyploidization: Regulation by Hippo and JNK Signaling and Conservation in Mammals. *PLoS One*. 2016 Mar 9;11(3):e0151251.

Nishimura T, Honda H, Takeichi M. Planar cell polarity links axes of spatial dynamics in neural-tube closure. *Cell*. 2012 May 25;149(5):1084-97.

Orr-Weaver TL. When bigger is better: the role of polyploidy in organogenesis. *Trends Genet*. 2015 Jun;31(6):307-15.

Zanet J, Freije A, Ruiz M, Coulon V, Sanz JR, Chiesa J, Gandarillas A. A mitosis block links active cell cycle with human epidermal differentiation and results in endoreplication. *PLoS One*. 2010 Dec 20;5(12):e15701.

REVIEWERS' COMMENTS:

Reviewer #1 (Remarks to the Author):

Kimura-Yoshida et al have performed significant additional work and addressed some of previous comments. Remaining, however, is my major concern about this work. I remain unconvinced that the work (the majority of the paper) on the embryoid bodies has relevance to normal development.

The authors show no clear in vivo correspondence to the highly abnormal, pathological-looking LM cells, a major focus of the paper. In fact, their new experiments show the artificiality of the system even more clearly as mis-expressing Grhl3 in early embryos leads to massively abnormal embryogenesis and early embryonic death. The authors claim (Fig. 2) that these embryos show multinucleated cells and increased actin but this is not convincing, and clearly no cells look anything similar to the LM cells. The same can be said for the later data (Fig. 6) that multinucleated Grhl3+ cells are found in the surface ectoderm. This is not convincing, and again the cluster of cells looks nothing like the LM cells. In sum, linking the work on embryoid bodies to surface ectoderm is not convincing. I doubt further work on these cells is fruitful, but if one were to pursue them, I believe looking to pathological formation of multinuclear cells would be most productive. For example, the authors might study autophagy and other mechanisms whereby such cells form. But I don't believe it will add valuable data to the understanding to the normal role of Grhl3.

Technical comments:

1. The data in Fig. 2n,o seems to show exclusive expression of EGFP and TROMA-1; not overlapping like the authors claim.
2. Supplementary Fig. 5 shows different Grhl3 location depending on the antibody without any explanation provided. In fact, antibodies against aa195-211 and aa478-493 are not clearly different than the background.
3. In the experiment in Fig. 7, the authors used antibody against aa478-493, which gave mostly cytoplasmic signal in Supplemental Figure 5, rather than antibody against aa49-134, which was the only antibody in Supplemental Figure 5 that gave a convincing signal over background and mainly gave nuclear staining.
4. The data on Rhogef19 in Figure 8 is not convincing. In the q-panel, the wholemount, there is no staining which is very surprising because Rhogef19 is expressed in mesoderm tissues as well and that signal should have been obvious. Also, in the sections I cannot see that the epidermis expression is clearly different in the surface ectoderm between the three genotypes (s-u).

Reviewer #2 (Remarks to the Author):

The authors have addressed all my concerns. As stated in my original review, this is a very interesting paper.

Reviewer #3 (Remarks to the Author):

The authors have satisfactorily addressed my comments in their revised manuscript. The new data strengthens the authors' hypothesis that cytoplasmic GRHL3 protein contributes to cell shape changes.

REF: NCOMMS-17-29972-B

“Cytoplasmic localization of GRHL3 upon epidermal differentiation triggers cell shape change for epithelial morphogenesis”

To the reviewers:

We would like to thank two reviewers, reviewer #2 and #3, for their supportive comments on our manuscript. We have addressed most of the concerns raised by reviewer #1 on a point-by-point basis.

Sincerely,

Isao Matsuo, Ph.D.

Chiharu Kimura-Yoshida, Ph.D.

Reviewer #1 (Remarks to the Author):

Kimura-Yoshida et al have performed significant additional work and addressed some of previous comments. Remaining, however, is my major concern about this work. I remain unconvinced that the work (the majority of the paper) on the embryoid bodies has relevance to normal development.

The authors show no clear in vivo correspondence to the highly abnormal, pathological-looking LM cells, a major focus of the paper. In fact, their new experiments show the artificiality of the system even more clearly as mis-expressing Grhl3 in early embryos leads to massively abnormal embryogenesis and early embryonic death. The authors claim (Fig. 2) that these embryos show multinucleated cells and increased actin but this is not convincing, and clearly no cells look anything similar to the LM cells. The same can be said for the later data (Fig. 6) that multinucleated Grhl3+ cells are found in the surface ectoderm. This is not convincing, and again the cluster of cells looks nothing like the LM cells. In sum, linking the work on embryoid bodies to surface ectoderm is not convincing. I doubt further work on these cells is fruitful, but if one were to pursue them, I believe looking to pathological formation of multinuclear cells would be most productive. For example, the authors might study autophagy and other mechanisms whereby such cells form. But I don't believe it will add valuable data to the understanding to the normal role of Grhl3.

Response:

Reviewer #1 still considers that LM-epidermal cells induced by *Grhl3* cDNA in *in vitro* cultures of embryoid bodies is artificial and abnormal as at the initial review. Additionally, the reviewer argues that there is no relevance between cells induced by *Grhl3* *in vitro* and *in vivo*. However, we are certain that LM-epidermal cells induced by *Grhl3* cDNA are neither artificial, nor abnormal, and that there is some relevance between LM-epidermal cells and cells *in vivo* induced by *Grhl3* for the following reasons:

i) Similarities between LM-epidermal cells from embryoid bodies and *in vivo* embryonic cells induced by *Grhl3*

Phenotypes of transgenic mice and electroporated cells mis-expressing *Grhl3* may not be completely identical with *in vitro* LM-epidermal cells, but clear similarities are present between *in vitro* and *in vivo* cells. In general, we understand that embryonic stem (ES) cells and

embryoid bodies are not identical to inner cell masses or epiblasts but they show significant similarities between *in vitro* and *in vivo*. We would like to emphasize that LM-epidermal cells *in vitro*, and transgenic or electroporated cells *in vivo* induced by *Grhl3* share some similarities. For example, multinucleation and the up-regulation of F-actin appear to be common, which have been explicitly demonstrated (Fig. 2g–u). Additionally, the expression of planar cell polarity (PCP) molecules was also up-regulated both *in vivo* and *in vitro* (Fig. 5a–d, 6u–x). These phenotypes further strengthen our assertion that several similar phenotypes are common to both LM-epidermal cells and *in vivo* cells mis-expressing *Grhl3*. So, we have added a paragraph discussing the limitations of our LM-epidermal cells *in vitro* model in Discussion of our re-revised manuscript.

In addition, the reviewer is also concerned about the very early embryonic lethality of *Grhl3* transgenic embryos. However, since *Grhl3* plays an important role in epidermal specification and morphogenesis, such as cell shape change for example, it is reasonable that *Grhl3*-misexpressing transgenic embryos show early embryonic lethality. Indeed, it would be rather irrational if *Grhl3* transgenic embryos survived beyond early embryonic stages. Consistent with this idea, when we mis-expressed *Grhl2* cDNA in transgenic mice, their embryos also showed embryonic lethality but were able to survive until the neurula stage (E9.5), showing defects of neural tube closure at the level of forebrain, and so forth (our unpublished observation, Attached data 1). The different phenotypes of *Grhl2* and *Grhl3* transgenic embryos clearly indicate that *Grhl3* has a unique role distinct from *Grhl2* during embryogenesis, which is also indicated by our findings of the inefficient induction of *in vitro* LM-epidermal cells with *Grhl1* and *Grhl2* cDNAs (Fig. 1 and the first submitted version of our manuscript). Thus, we consider that these findings further support the crucial role of *Grhl3* in *in vivo* normal development.

ii) Technical usefulness of formation of LM-epidermal cells

The usefulness of a culture system of embryoid bodies is generally and widely accepted (Simunovic et al., 2017). A lot of differentiation systems derived from ES cells have been developed through embryoid bodies. Thus, these *in vitro* systems cannot be considered as artificial. It is noted that by exploiting such *in vitro* LM-epidermal cells, we were able to identify proteins that interacted with GRHL3 and by further analyzing the function of these proteins we now have promising molecules mediating GRHL3 translocation *in vitro* (our unpublished observations). Taken together, since this is the first report regarding the differentiation of LM-epidermal cells from embryoid bodies, the induction of LM-epidermal cells itself deserves to be reported in terms of the technical benefits.

In addition, quite a few assay systems are used to investigate the activity of non-canonical Wnt signaling. However, the activation of non-canonical Wnt or PCP activation is essential to the formation of LM-epidermal cells. Thus, if our methods are reported, many researchers who are studying non-canonical Wnt and PCP signaling, and epidermal differentiation may use our unique differentiation system.

Another benefit to using *in vitro* LM-epidermal cells is for animal welfare and ethical reasons. The replacement of live animal experiments with *in vitro* experiments are strongly expected by the public as well as in scientific fields and has now become topical. In fact, it may be possible to analyze enormous animals instead of embryoid bodies in *in vitro* culture. However, from an animal welfare viewpoint, it is preferred that an *in vitro* cell culture system be used as much as possible. In fact, our national and institutional guidelines on the use of

laboratory animals strongly call for the preferential use of *in vitro* experimental systems, where possible. *Nature Communications* also endorses Animal Research: Reporting of In Vivo Experiments (ARRIVE) guidelines (Kilkenny et al., 2010). Thus, in our current manuscript, we have carefully designed animal experiments so as to reduce their number. Thanks to the *in vitro* experiments of Fig. 3 and 4, we were able identify molecular mechanisms of LM-epidermal formation through embryoid bodies, i.e. cytoplasmic localization of nucleus GRHL3 involves activation of non-canonical Wnt signaling. This mechanism was ultimately supported and confirmed by a limited and rational number of *in vivo* animal experiments (Fig. 6–8).

iii) Normal but not pathologic characteristics of cells induced by *Grhl3*.

The reviewer argues that cellular phenotypes in response to the mis-expression of *Grhl3* are pathologic and related to autophagy. We are uncertain whether these *Grhl3*-expressing cells can be directly linked to autophagy. Regardless, it is generally considered that autophagy is neither an abnormal nor artificial phenomenon but rather an essential process in normal development (Boya et al., 2018; Hale et al., 2013). Additionally, it is well known that the enlargement of cell and organ sizes is not pathologic but rather very important for normal developmental processes (Anzi et al., 2018; Patra et al., 2018; Yu et al., 2015; Zhao et al., 2011). Thus, we believe that cellular phenotypes provided by the mis-expression of *Grhl3* are not pathologic but rather reflect some aspects of normal embryonic development.

In addition, we have already intensively characterized LM-epidermal cells as shown in Fig. 1 and Supplementary Fig. 1, and clearly found that LM-epidermal cells have the characteristics of more mature epidermal cells. Moreover, all reviewers would now agree with the notion that non-canonical Wnt signaling or PCP molecules are involved in the formation of LM-epidermal cells. We also consider that these unique characteristics involving specific signaling seem to reflect normal developmental processes but not pathologic phenomena.

Taken together, we consider that the formation of LM-epidermal cells through embryoid bodies is very consistent with normal development and this methodology is scientifically and ethically rational.

Anzi S, et al. Postnatal exocrine pancreas growth by cellular hypertrophy correlates with a shorter lifespan in mammals. *Dev Cell*. 2018 Jun 18;45(6):726-737.

Boya P, Codogno P, Rodriguez-Muela N. Autophagy in stem cells: repair, remodelling and metabolic reprogramming. *Development*. 2018 Feb 26;145(4). dev146506.

Hale AN, Ledbetter DJ, Gawriluk TR, Rucker EB 3rd. Autophagy: regulation and role in development. *Autophagy*. 2013 Jul;9(7):951-972.

Kilkenny C, et al. Improving bioscience research reporting: the ARRIVE guidelines for reporting animal research. *PLoS Biol*. 2010 Jun 29;8(6):e1000412.

Patra KC, Bardeesy N. A cell size theory of aging. *Dev Cell*. 2018 Jun 18;45(6):665-666.

Simunovic M, Brivanlou AH. Embryoids, organoids and gastruloids: new approaches to understanding embryogenesis. *Development*. 2017 Mar 15;144(6):976-985.

Yu FX, Zhao B, Guan KL. Hippo pathway in organ size control, tissue homeostasis, and cancer. *Cell*. 2015 Nov 5;163(4):811-828.

Zhao B, Tumaneng K, Guan KL. The Hippo pathway in organ size control, tissue regeneration and stem cell self-renewal. *Nat Cell Biol*. 2011 Aug 1;13(8):877-883.

Technical comments:

1. The data in Fig. 2n,o seems to show exclusive expression of EGFP and TROMA-1; not overlapping like the authors claim.

Response:

The data in Fig. 2n,o show merged confocal images after electroporation of solely *EGFP* cDNA without *Grhl3* cDNA. Therefore, these two panels indicated the rationality of our electroporation system to confirm whether *EGFP* cDNA was correctly electroporated into the surface region of the embryo. In order to avoid a misunderstanding, we are able to show each confocal image of EGFP, TROMA-1 and DAPI, differently, instead of Fig. 2n,o (Attached Data 2). It can be clearly noticed and agreed upon that EGFP expression overlaps with TROMA-1 positive regions; however, these two expressions are not exclusive.

2. Supplementary Fig. 5 shows different *Grhl3* location depending on the antibody without any explanation provided. In fact, antibodies against aa195-211 and aa478-493 are not clearly different than the background.

3. In the experiment in Fig. 7, the authors used antibody against aa478-493, which gave mostly cytoplasmic signal in Supplemental Figure 5, rather than antibody against aa49-134, which was the only antibody in Supplemental Figure 5 that gave a convincing signal over background and mainly gave nuclear staining.

Response:

As we have shown in Supplementary Fig. 5, immunohistochemistry with antibodies against aa195-211 and aa478-493 of GRHL3 in wild-type embryos during neural tube closure indicates that GRHL3 is expressed mainly at the cell surface and cytoplasm of lateral surface ectoderm and partly in the nucleus of neural folds (Supplementary Fig. 5c,e). As for immunohistochemistry using these two antibodies in *Grhl3*^{cre/cre} mutant embryos, some non-specific background staining seems to be present but specific staining, which is found in the wild-type embryos, is convincingly lost with both antibodies (Supplementary Fig. 5d,f).

Since the aa478-493 antibody displayed less background staining, we decided to use this antibody to analyze the distribution of GRHL3 in *Grhl3*^{NLS/NLS} mutant embryos (Fig. 7h-k). Since the aa49-134 antibody recognizes nuclear localized GRHL3 preferentially, this antibody may not recognize cytoplasmic GRHL3 correctly. So, we concluded that the aa478-493 antibody was appropriate and suitable to detect the translocation of GRHL3 from the cytoplasm into nucleus, i.e. it is easier to detect how much cytoplasmic GRHL3 is reduced or not.

To avoid the confusion, we have added these above statements in Methods and figure legends of Supplementary Figure 5 in the re-revised version of the manuscript.

4. The data on *Rhogef19* in Figure 8 is not convincing. In the q-panel, the wholemount, there is no staining which is very surprising because *Rhogef19* is expressed in mesoderm tissues as well and that signal should have been obvious. Also, in the sections I cannot see that the epidermis expression is clearly different in the surface ectoderm between the three genotypes (s-u).

Response:

In line with this comment, we have attached additional sectional views as shown below. These data clearly show that *Rhogef19* is expressed in mesodermal and endodermal tissues in wild-type, *Grhl3*^{cre/cre}, and *Grhl3*^{NLS/NLS} embryos in a similar manner (Attached data 3 A-A'',C-C''). However, *Rhogef19* expression in the surface ectoderm is severely reduced in *Grhl3*^{cre/cre}, but not in wild-type and *Grhl3*^{NLS/NLS} embryos (Attached data 3 B-B'').

Reviewer #2 (Remarks to the Author):

The authors have addressed all my concerns. As stated in my original review, this is a very interesting paper.

We would like to thank the Reviewer for her/his support of publication on our paper.

Reviewer #3 (Remarks to the Author):

The authors have satisfactorily addressed my comments in their revised manuscript. The new data strengthens the authors' hypothesis that cytoplasmic GRHL3 protein contributes to cell shape changes.

We would like to thank the Reviewer for her/his support of publication on our paper.

Attached data 1

E8.5

Attached data 2

Attached data 3